# Non-coplanar helimagnetism in the layered van-der-Waals metal DyTe₃

Shun Akatsuka[1,9], Sebastian Esser [1,9] ✉, Shun Okumura [1], Ryota Yambe[1], Rinsuke Yamada [1], Moritz M. Hirschmann [2], Seno Aji [3,8], Jonathan S. White [4], Shang Gao[5], Yoshichika Onuki[2], Taka-hisa Arima [2,6], Taro Nakajima [2,3] & Max Hirschberger [1,2,7] ✉

Van-der-Waals magnetic materials can be exfoliated to realize ultrathin sheets or interfaces with highly controllable optical or spintronics responses. In majority, these are collinear ferro-, ferri-, or antiferromagnets, with a particular scarcity of lattice-incommensurate helimagnets of defined left- or right-handed rotation sense, or helicity. Here, we report polarized neutron scattering experiments on DyTe₃, whose layered structure has highly metallic tellurium layers separated by double-slabs of dysprosium square nets. We reveal cycloidal (conical) magnetic textures, with coupled commensurate and incommensurate order parameters, and probe the evolution of this ground state in a magnetic field. The observations are well explained by a one-dimensional spin model, with an off-diagonal on-site term that is spatially modulated by DyTe₃'s unconventional charge density wave (CDW) order. The CDW-driven term couples to antiferromagnetism, or to the net magnetization in an applied magnetic field, and creates a complex magnetic phase diagram indicative of competing interactions in this easily cleavable van-der-Waals helimagnet.

Magnetism in layered materials, held together by weak van-der-Waals interactions, is an active field of research spurred on by the discovery of magnetic ordering in monolayer sheets of ferromagnets and antiferromagnets[1–4]. At the frontier of this field, helimagnetic layered systems, where magnetic order has a fixed, left- or right-handed rotation sense, have been predicted to host complex spin textures[5,6] and to serve as controllable multiferroics platforms, where magnetic order is readily tuned by electric fields or currents[7–10]. However, most layered van-der-Waals magnets are commensurate ferro-, antiferro-, or ferrimagnets[3,4]; the rare helimagnets provided to us by nature are often modulated along the stacking direction, with relatively simple spin arrangement in individual layers (Supplementary Table I).

In the quest for helimagnetism in layered structures with weak van-der-Waals bonds, we focus on rare earth tritellurides $R$Te₃ ($R$: rare earth element). These materials form a highly active arena of research regarding the interplay of correlations and topological electronic states[11–15]. Their structure, which can be exfoliated down to the thickness of a few monolayers[16,17], is composed of tellurium Te₂ double-layers and covalently bonded $R$Te slabs, with characteristic square net motifs in both (Fig. 1a)[18]. Tellurium $5p$ electrons are localized in Te₂ square net bilayers ($ac$ plane), in which they form dispersive bands with elevated Fermi velocity[16,19–21]. This quasi two-dimensional electronic structure amplifies correlation phenomena, such as the formation of charge density wave (CDW) order[22] and superconductivity[23,24],

¹Department of Applied Physics, The University of Tokyo, Bunkyo-ku, Tokyo 113-8656, Japan. ²RIKEN Center for Emergent Matter Science (CEMS), Wako, Saitama 351-0198, Japan. ³The Institute for Solid State Physics, The University of Tokyo, Kashiwa 277-8581, Japan. ⁴Laboratory for Neutron Scattering and Imaging (LNS), Paul Scherrer Institute (PSI), 5232 Villigen, Switzerland. ⁵Department of Physics, University of Science and Technology of China, Hefei 230026, China. ⁶Department of Advanced Materials Science, The University of Tokyo, Kashiwa 277-8561, Japan. ⁷Quantum-Phase Electronics Center (QPEC), The University of Tokyo, Bunkyo-ku, Tokyo 113-8656, Japan. ⁸Present address: Department of Physics, Faculty of Mathematics and Natural Sciences, Universitas Indonesia, Depok 16424, Indonesia. ⁹These authors contributed equally: Shun Akatsuka, Sebastian Esser. ✉e-mail: esser@g.ecc.u-tokyo.ac.jp; hirschberger@ap.t.u-tokyo.ac.jp

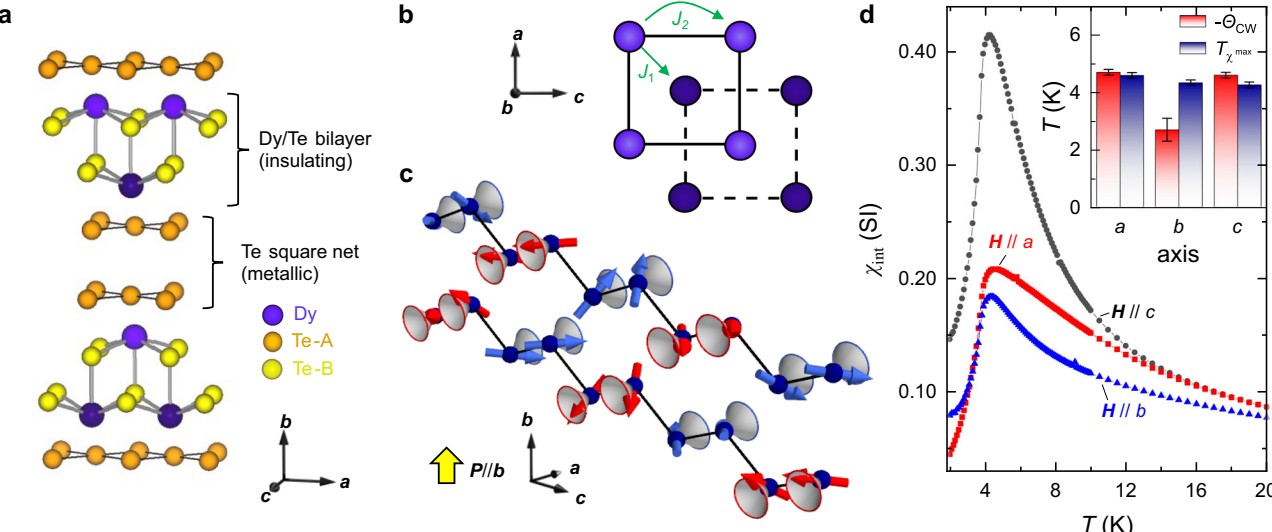

**Fig. 1 | Conical helimagnetism in the layered square-lattice antiferromagnet DyTe₃. a** Crystallographic unit cell with covalently bonded DyTe bilayers and metallic Te bilayers. **b** Magnetic exchange interactions in a single DyTe double-square net bilayer. **c** Zigzag chain illustration of double-square net structure in DyTe₃. The interactions $J_1$, $J_2$ connect nearest and next-nearest neighbors in the zigzag chain model, respectively; but the inter-atomic distance is shorter for $J_2$, and its antiferromagnetic coupling strength is dominant. Conical, noncoplanar helimagnetism is resolved in the zero-field ground state by neutron scattering. The cone direction, parallel to the $a$-axis, alternates both between pairs of magnetic sites in a zigzag chain, and between stacked zigzag chains. This texture causes polarization along the $b$-axis, i.e., perpendicular to the square net bilayers (yellow arrow). The full magnetic unit cell extends two times further along the chain direction (Supplementary Fig. 1). **d** Weak anisotropy of the magnetic susceptibility $\chi$ in DyTe₃. The softest direction is $\mathbf{H} \| c$, consistent with the modulation direction of the magnetic order in (**c**). The inset shows Curie–Weiss temperatures and temperatures $T_\chi$ of maximal $\chi$, measured in a small magnetic field along three crystallographic directions. Error bars correspond to statistical uncertainties of Curie–Weiss fitting and temperature sampling width, respectively.

while also hosting protected band degeneracies[16,25]. In view of intense research efforts on $R$Te₃, it is remarkable that their magnetism has never been discussed in detail; in particular, no full refinement of magnetic structures is available[26–31].

Here, we report on helimagnetic, cone-type orders of DyTe₃ using polarized elastic neutron scattering. We reveal the magnetic texture in real space, probe its evolution with temperature and magnetic field, and discuss its relationship to CDW formation. In DyTe₃, dysprosium moments are arranged in square net bilayers, where each ion has neighbors within its own layer, and within the respective other layer (Fig. 1b). As all magnetic orders of DyTe₃ observed here are uniform along the crystallographic $a$-axis, it is reasonable to understand each square net bilayer as an effective zigzag chain of magnetic rare earth ions and to define magnetic interactions $J_1$ and $J_2$ in terms of nearest- and next-nearest neighbors on the zigzag chain, respectively. On such chains, our experiment shows that pairs of ions have cones pointing along the same direction, followed by a flip of the cone axis (Fig. 1c, which illustrates half a magnetic unit cell). The coupling between two DyTe bilayers, i.e., between two zigzag chains, is antiferromagnetic. Despite this complex cone arrangement, the magnetic structure defines a fixed sense of rotation, or helicity. Our theoretical spin model shows that CDW order in rare earth tellurides causes local symmetry breaking and drives lattice-incommensurate magnetism when combined with antiferromagnetic interactions or with a net magnetization. We further discuss magneto-crystalline anisotropy in this layered structure, with an unconventional combination of metallic and covalent bonds. Helimagnetism of Dy rare earth moments with $4f^9$ magnetic shell emerges despite the naive expectation of easy-axis or easy-plane anisotropy for $^{2S+1}L_J = {}^6H_{15/2}$, with large orbital angular momentum $L = 5$.

## Results
### Magnetic properties of DyTe₃
Some essential magnetic properties of DyTe₃ are apparent already from the magnetic susceptibility $\chi$ in Fig. 1d. In the high temperature regime, anisotropy in the Curie–Weiss law indicates easy-plane behavior of magnetic moments, favouring the $ac$ plane with uniaxial anisotropy constant $K_1 = 22(1)$ kJ m⁻³, see "Methods". At low temperatures, the strongest enhancement of $\chi$ occurs when the magnetic field $\mathbf{H}$ is along the $c$-axis, i.e., parallel to the zigzag direction defined in Fig. 1c. We deduce that the magnetic moments are aligned, predominantly, along the $a$- and $b$-axes. All susceptibility curves show maxima around $T_\chi = 4.5$ K, quite far above the onset of three-dimensional, long-range magnetic order, as shown in the following.

We characterize the phase transition in DyTe₃ using thermodynamic and transport probes in Fig. 2a, b. The specific heat $C(T)$ shows a two-peak anomaly, describing the transitions from the paramagnetic regime to Phase II at $T_{N2} = 3.85$ K and to Phase I at $T_{N1} = 3.6$ K. Below $T_{N1}$, the resistivities in the $ac$ basal plane drop abruptly, suggesting a clear correlation between the behavior of freely moving conduction electrons and the magnetic structure. The presence of a partial charge gap in the electronic structure, related to magnetic ordering, is inferred from an increase of the ratio of resistivities $\rho_a$ and $\rho_c$. Simultaneously, as discussed in the following, strong neutron scattering intensity appears below $T_{N2}$ at two independent positions in reciprocal space, c.f. Fig. 2c. The magnetic scattering intensity rises abruptly upon cooling below $T_{N2}$.

To obtain this neutron data, a single-domain crystal of DyTe₃ is mounted on an aluminum holder and is pre-aligned by means of Laue X-ray diffraction. More quantitatively, we determine the crystallographic directions in DyTe₃ using the crystallographic extinction rule (Supplementary Fig. 11). Figure 2d describes the geometry of our neutron scattering experiment. The scattering plane that includes the incoming and outgoing neutron beams $\mathbf{k}_i$ and $\mathbf{k}_f$, is spanned by the $b$- and $c$-axes. Hence, reflections of the type $\mathbf{Q} = \mathbf{k}_f - \mathbf{k}_i$ with Miller indices ($0KL$) can be detected, as in Fig. 2e, where a line scan along ($01L$) provides sharp magnetic intensity. Several types of magnetic peaks $\mathbf{Q} = \mathbf{G} + \mathbf{q}$, with $\mathbf{G}$ a reciprocal lattice vector, are observed: primarily, a commensurate (C) reflection $\mathbf{q}_{AFM} = (0, b^*, q_{AFM})$, $q_{AFM} = 0.5c^*$ and an incommensurate (IC) reflection $\mathbf{q}_{cyc} = (0, b^*, q_{cyc})$, $q_{cyc} = 0.207\,c^*$, where $b^* = 2\pi/b$ and $c^* = 2\pi/c$ are reciprocal lattice constants, see "Methods".

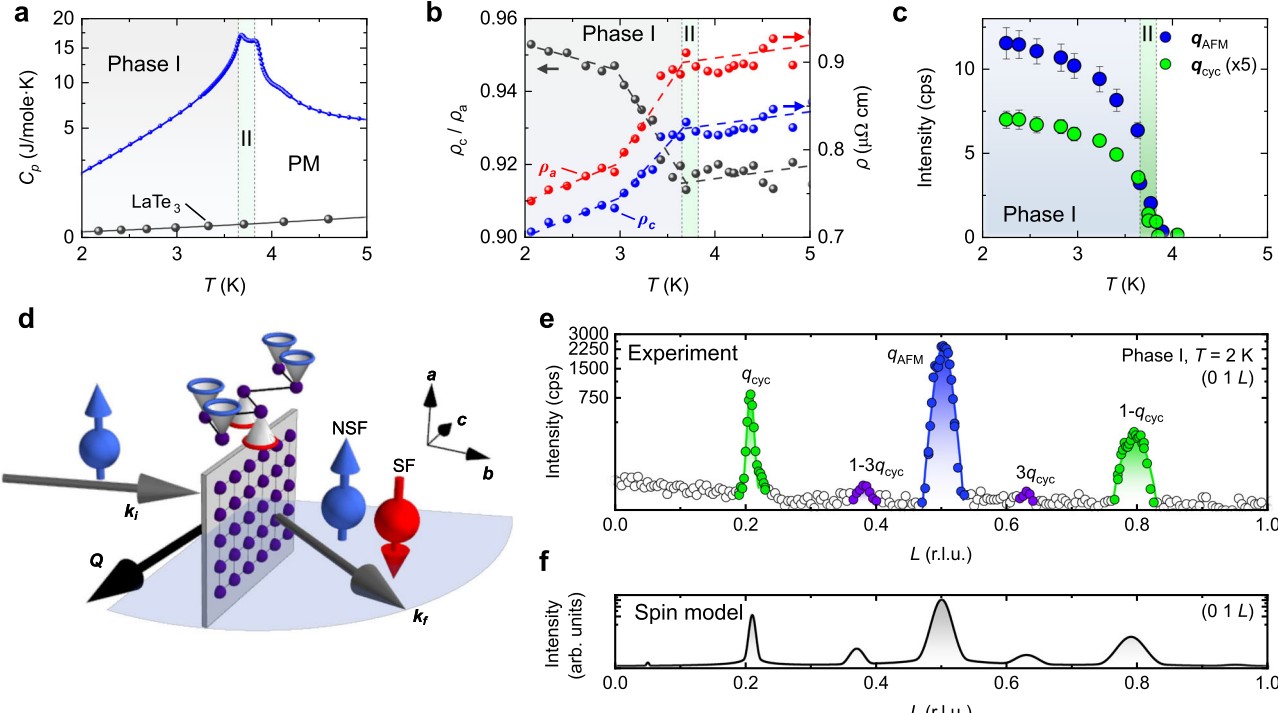

**Fig. 2 | Bulk characterization and zero-field magnetic structure of DyTe₃ from elastic neutron scattering. a** Specific heat showing two transitions at $T_{N1}$ and $T_{N2}$, with reference data from the nonmagnetic analog LaTe₃[32]. **b** Below $T_{N1}$, where resistivity in the $ac$ basal plane drops sharply, significant anisotropy $\rho_c/\rho_a$ indicates (partial) gapping of electronic states along the $c$-axis. **c** Temperature dependence of magnetic scattering intensity for antiferromagnetic $\mathbf{q}_{AFM}$ and cycloidal $\mathbf{q}_{cyc}$ reflections, with Miller indices $(HKL) = (0, 1, 0.5)$ and $(0, 1, 0.207)$, respectively. Gray and green background shading mark Phases I and II, respectively. Error bars correspond to statistical uncertainties of Gaussian fits to the magnetic reflections.

**d** Experimental geometry for polarized neutron scattering from layered DyTe₃ (gray plane, square net of Dy indicated). The scattering plane (blue) is spanned by wavevectors $\mathbf{k}_i$, $\mathbf{k}_f$ of incoming and outgoing neutron beams, respectively. Separating spin flip (SF, red) and non-spin flip (NSF, blue) scattering intensities at the detector, we identify conical magnetic order (top). **e** Linescan in momentum space, with highlights for three types of reflections including a weak higher harmonic corresponding to $3 \times \mathbf{q}_{cyc}$. **f** Simulation of magnetic intensity from magnetic structure model, corresponding to the linescan in (**e**), see "Methods".

There is also a higher harmonic (3Q) reflection, corresponding to three times the length of $\mathbf{q}_{cyc}$, which describes an anharmonic distortion of the texture. As a main result of this work, we ascribe $\mathbf{q}_{cyc}$ to a cycloidal structure in the magnetic ground state of DyTe₃, that results from a coupling $q_{AFM} \pm q_{CDW}$ between $\mathbf{q}_{AFM}$ and a CDW modulation $\mathbf{q}_{CDW}$ at $(0, 0, q_{CDW})$, $q_{CDW} = 0.29\,c^*$ in the rare earth tritelluride family.

## Ground state magnetic structure model

We reveal the helimagnetic structure in the ground state of DyTe₃ using polarized neutron scattering. As shown in Fig. 2d, the incident neutron spins are polarized perpendicular to the scattering plane. We employ a magnetized single-crystal analyzer to select the energy and spin state of the scattered neutrons, see "Methods". The scattering processes in which the neutron spins are reversed (remain unchanged) is referred to as spin-flip, SF (non-spin-flip, NSF). For SF scattering, it is required that the magnetic moments **m** have a component perpendicular to the spin of the incoming neutron. This means that SF and NSF scattering detect components of **m** within $(m_b, m_c)$ and perpendicular to $(m_a)$ the scattering plane, respectively.

Polarization analysis of the magnetic reflections shows that $\mathbf{q}_{cyc}$ and $\mathbf{q}_{AFM}$ relate to different vector components of the ordered magnetic moment (Fig. 3a, b, e, f). We find no hint of SF scattering at $\mathbf{q}_{AFM}$, demonstrating collinear antiferromagnetism with magnetic moments exclusively along the $a$-direction. The incommensurate part $\mathbf{q}_{cyc}$, in contrast, has no NSF intensity and roughly equal SF signals at various positions in reciprocal space (Fig. 3e, f and insets). As neutron scattering detects the part of **m** that is orthogonal to **Q**, comparison of magnetic reflections situated at nearly orthogonal directions in

momentum space suggests $m_b$ and $m_c$ components are both finite in the ground state.

We determine the quantitative relationship between magnetic moments within a DyTe bilayer (within an effective zigzag chain), by comparing the observed and calculated magnetic structure factors under the constraints imposed by polarized neutron scattering, c.f. Supplementary Fig. 8. The analysis for $\mathbf{q}_{AFM}$ demonstrates up–up–down–down type ordering along the zigzag chain, visualized from two perspectives in Fig. 3c, d. At $\mathbf{q}_{cyc}$, the refinement yields a cycloid with a phase delay $\delta$ between the upper and lower sheets in a zigzag chain, see Fig. 3g. In effect, pairs of nearly parallel magnetic moments are followed by a significant rotation of the moment direction. The coupling between zigzag chains is antiferromagnetic, as imposed by the Miller index $K = 1$ ($q_b = b^*$) in both $\mathbf{q}_{cyc}$ and $\mathbf{q}_{AFM}$. Combining the three components $m_a$, $m_b$, and $m_c$, we realize the noncoplanar, helimagnetic cone texture of Fig. 1c that is, to our knowledge, unique in both insulators and metals. In Supplementary Sections III A, IV A, we discuss the presence of magnetic domains in the sample and how the occurrence of higher harmonic reflections further supports our magnetic structure model.

## Charge density wave and magnetic order

We now argue that cone-type magnetism in DyTe₃ is realized through (i) coupling of the magnetic texture to CDW order and (ii) unconventionally weak single-ion anisotropy. We turn first to (i), that is the role of the CDW in stabilizing noncoplanar helimagnetism in DyTe₃. We use a 1D chain model to reproduce key features of the modulated magnetic order, neglecting the material's three-dimensionality, see "Methods".

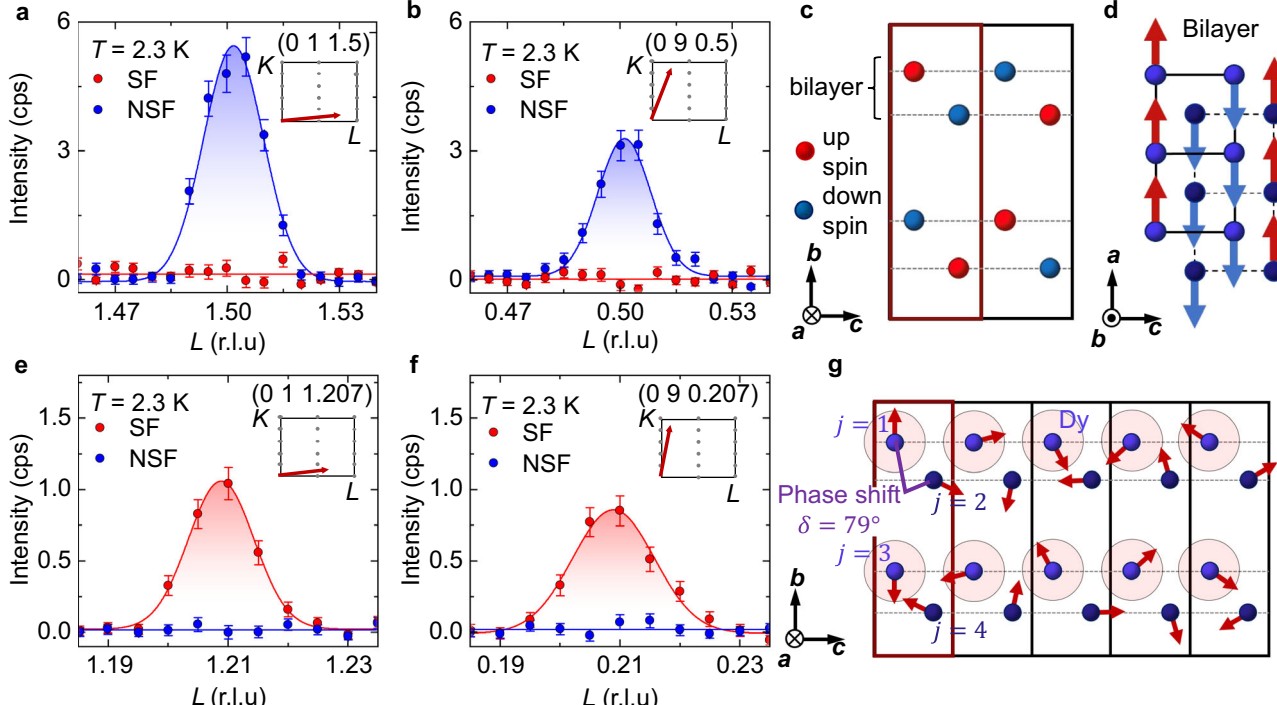

**Fig. 3 | Two spin components of the cone-type order in the ground state of DyTe₃ (Phase I).** In the geometry of Fig. 2d, non-spin-flip (NSF) and spin-flip (SF) neutron scattering measure the magnetic moment along the crystallographic $a$-axis and in the $bc$ plane, respectively. **a, b** Polarization analysis of antiferromagnetic component $q_{AFM}$: absence of spin flip (SF) intensity indicates absence of $m_b$ and $m_c$, while non-spin flip (NSF) intensity reveals dominant $m_a$. **c, d** Two views of the antiferromagnetic collinear component derived from this data, confirmed by full refinement of a number of magnetic reflections in Supplementary Fig. 8. Only Dy atoms are shown. **e, f** Polarization analysis for the incommensurate cycloidal reflection $q_{cyc}$. **g** Derived magnetic structure model for $q_{cyc}$, where $m_a$ (NSF)

vanishes while $m_c$ and $m_b$ (SF) both appear at comparable magnitudes in Phase I. The index $j$ labels four dysprosium atoms in four layers within the chemical unit cell (red box), where the cycloids at $j = 2, 4$ are phase-shifted with respect to cycloids at $j = 1, 3$. The numerical value of $\delta$ is determined in Supplementary Section VI. As for **a, b, e, f** we have subtracted background signals, and corrected the effect of imperfect beam polarization. The error bars correspond to Poisson counting errors of the integrated neutron scattering intensity, given in counts per second (cps). Each inset depicts the orientation of the measured **Q**-vector within the (0$KL$) reciprocal space.

In DyTe₃, the local environment and bond characteristics of dysprosium ions in a DyTe square net bilayer (in a zigzag chain) are spatially modulated by the CDW in the adjacent Te₂ sheets, c.f. Fig. 4b[11–14,18,22–24,31–33]. The simplest model approach is to decouple the zigzag chain, with two atoms per unit cell, into two one-dimensional chains, with one atom per unit cell. This allows for a two-parameter model, built from Ising-like exchange interactions together with a spatially modulated on-site coupling,

$$\mathcal{H} = \sum_n \left[ J_2^{AFM} S_n^a S_{n+1}^a - E_{CDW}^{ab} \cos(q_{CDW} z_n) S_n^a S_n^b - E_{CDW}^{ac} \sin(q_{CDW} z_n) S_n^a S_n^c \right]$$

(1)

where $n$ counts magnetic sites, e.g., on the upper half of the zigzag chain. The $z_n$ are spatial positions along the zigzag chain ($c$-axis). All the coupling constants – $J_2^{AFM}$, $E_{CDW}^{ab}$, and $E_{CDW}^{ac}$ – are positive. The $E_{CDW}^{ab}$ and $E_{CDW}^{ac}$ terms are allowed by global and local mirror symmetry breaking due to the CDW, respectively. We may also introduce a Zeemann term to explain the behavior in a magnetic field and further inter-chain coupling to connect the two chains (Supplementary Section I).

This 1D model naturally creates different modulation period for the $a$ and $bc$ spin components and robustly reproduces two types of magnetic reflections, $q_{AFM}/c^* = 0.5$ and $q_{cyc}/c^* = 0.5 - 0.293 = 0.207$. In good consistency with the experiment, Fig. 4c shows that $I_{cyc}$ on the order of 10% of $I_{AFM}$ can be induced within this model. Note that, in addition to the magnetic texture in Fig. 1c, alternative (out-of-phase) locking between cycloid and antiferromagnetic component is also possible (Supplementary Fig. 1). Based on scattering techniques alone,

it is not possible to decide the phase-shift between CDW and the spin cycloid, and between the antiferromagnetic and incommensurate components of the magnetic order.

## Weak magneto-crystalline anisotropy
While strongly anisotropic magnetism is naively expected for dysprosium's $^6H_{15/2}$ shell, we here report a conical state with comparable $m_a, m_b, m_c$ in DyTe₃. Consider the local environment of a single Dy in Fig. 4d: Te-B ions form covalent bonds with the central Dy, while the point charges of Te-A are effectively screened by itinerant electrons in the conducting tellurium slab. We model the sequence of crystal electric field (CEF) states for the $4f^9$ shell of dysprosium as a function of the effective crystal field charge $c$ situated on Te-A and Te-B ions (Supplementary Fig. 15). Figure 4e illustrates two limiting cases: when Te-A and Te-B contribute equally to the CEF, the $4f^9$ charge cloud is compressed along the $b$-direction, with $|J_b = \pm 15/2\rangle$ dominating the ground state wavefunction, and with effective out-of-plane magnetic anisotropy. Likewise, zero contribution of Te-A, i.e., highly efficient metallic screening of CEFs, favors the prolate orbital $|J_b = \pm 1/2\rangle$ with easy-axis anisotropy.

Adding exchange interactions $E_{ex}$ as an effective magnetic field, the CEF Hamiltonian of a point charge model is diagonalized for the orthorhombic environment of Dy, see "Methods". The resulting free energy density described by two parameters $K_1 \cos^2(\theta) + K_2 \sin^2(\theta) \cos^2(\phi)$, where $\theta, \phi$ are spherical coordinates with respect to the $b$ and $c$ crystal axes, respectively. Figure 4f, g testifies to a transition from easy-axis to easy-plane anisotropy through a sign change of $K_1$ at intermediate charge ratio (pink line). Two green lines

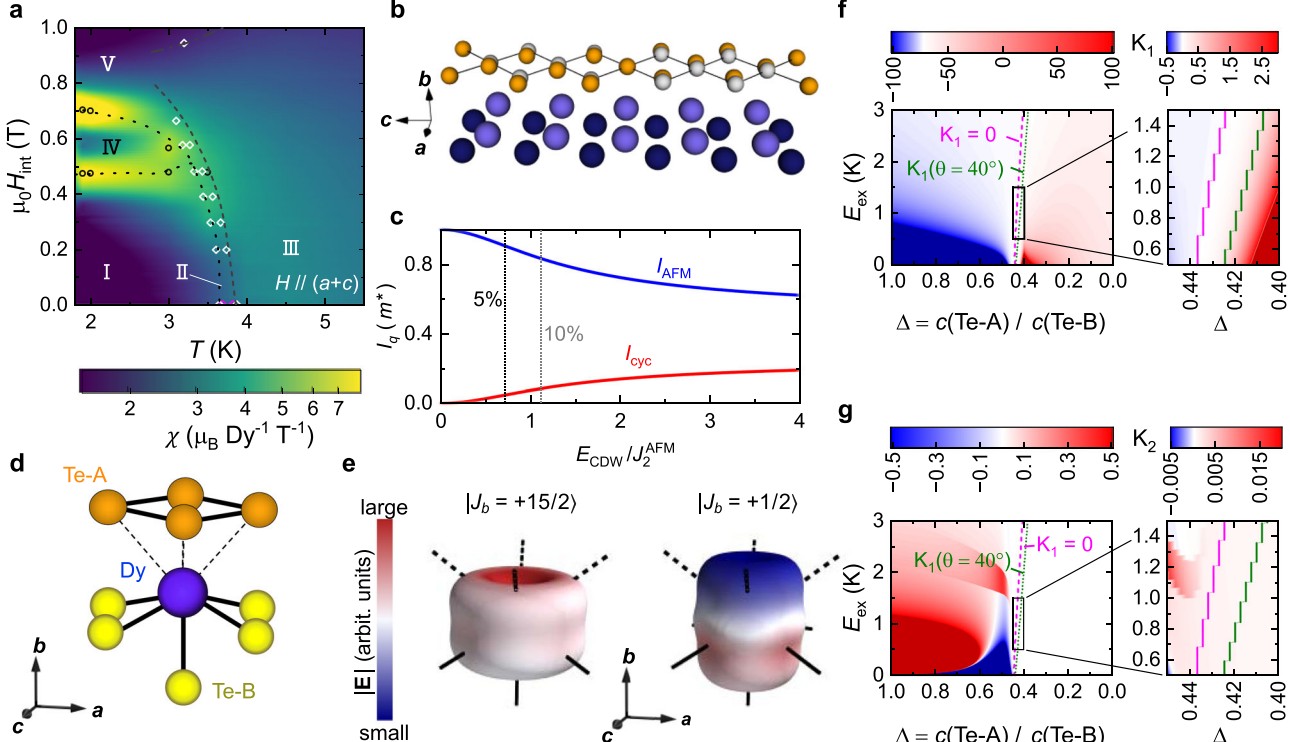

**Fig. 4 | Modeling helimagnetism in DyTe₃. a** Magnetic susceptibility $\chi$ for **H**||[101], overlaid with phase boundaries from specific heat (white open circles) and magnetization (black circles): We label phases I (conical ground state), II, III, IV, and V. **b** CDW on tellurium layers[18], where orange (gray) spheres are distorted (undistorted) ionic positions. Coupling of Dy (violet) and CDW drives lattice-incommensurate magnetic order through spatially modulated distortion of the local environment of Dy-ions. **c** Squared moment amplitudes $I_{AFM} = |S^a(q = 0.5c^*)|^2$ (blue) and $I_{cyc} = |S^b(q = 0.5c^* \pm q_{CDW})|^2 + |S^c(q = 0.5c^* \pm q_{CDW})|^2$ (red) as functions of $E_{CDW}/J_2^{AFM}$, where $E_{CDW} = E_{CDW}^{ab} = E_{CDW}^{ac}$, from model calculations according to Eq. (1); five and ten percent threshold for the incommensurate part indicated by black and gray dashed lines, see "Methods". **d** Local environment of Dy in DyTe₃, with

covalent (metallic) bonds to Te-B (Te-A) depicted by solid (dashed) lines, respectively. **e** Charge density (CD) of Dy $4f^9$ shell under the influence of crystal electric fields (CEF) from the surrounding ions, with exaggerated non-spherical part. If metallic and covalent bonds cause CEF of roughly equal strength (if metallic bonds are screened), oblate $|J_b = \pm15/2\rangle$ (prolate $|J_b = \pm1/2\rangle$) is the lowest energy CEF doublet. This favors out-of-plane (in-plane) magnetization. Color on CD isosurfaces indicates amplitude |**E**| of the local CEF. **f, g** Anisotropy constants $K_1$ and $K_2$ calculated for $4f^9$ multiplet in DyTe₃, see "Methods". The abscissa describes the relative weight of CEF charges $c$ on Te-A and Te-B sites. Green lines bound an intermediate regime of weak $K_1$, where spin tilting and conical order are allowed.

bound the regime where easy-axis (easy-plane) anisotropy is not strong enough to prevent tilting of **m** along directions intermediate between $b$-axis and the $ac$-plane. Constraining $E_{ex}$ in agreement with $T_{N2}$ and requiring easy-plane anisotropy $K_1 > 0$, we identify the black box in Fig. 4f, g to capture a parameter range well consistent with experiment. Here, the model yields $K_2 > 0$, meaning $m_a$ is preferred over $m_c$.

## Magnetic phase diagram and small-angle neutron scattering

We are ready, now, to consider the evolution of magnetic order in DyTe₃ as a function of temperature and magnetic field. Figure 4a shows a contour map of the magnetic susceptibility $\chi$, see "Methods", where the external magnetic field is applied along the in-plane direction [101], i.e., **H**||$(a + c)$. Heating the sample above $T_{N1} = 3.6$ K in zero field, we observe a peak splitting of **q**$_{AFM}$, and a concomitant shift in **q**$_{cyc}$ that indicates the sustained coupling of the two ordering vectors, via the CDW, at elevated temperatures (Supplementary Fig. 13). The sharp enhancement of $\chi_c$ in Fig. 1a further suggests that $m_a$, $m_b$ survive to higher temperature than $m_c$, consistent with a putative incommensurate, fan-like order in Phase II, which warrants further study.

To explore the high-field regime, confirm the coupling between **q**$_{AFM}$ and **q**$_{cyc}$ in Phase I, and investigate the correlation of CDW and magnetic order, we carried out small-angle neutron scattering (SANS) experiments in a magnetic field. Figure 5a describes the geometry of our SANS experiment and Fig. 5b shows the obtained zero field $(0, K, L)$ map, while Fig. 5c reduces the map into principal line cuts. The non-

coplanar, helimagnetic cone texture with coupled **q**$_{AFM}$ and **q**$_{cyc}$ is stable up to $\mu_0 H = 0.5$ T for **H**||$c$, c.f. Fig. 5f. In fact, these data further support the existence of coupled commensurate and incommensurate order parameters in Phase I, and help to exclude a domain separation scenario (Supplementary Fig. 14). A pair of phase transitions to Phases IV and V is visualized in Fig. 5f and Supplementary Fig. 14. In Phase V, c.f. Fig. 5d, e, which is realized when the external magnetic field exceeds $\mu_0 H = 0.7$ T, strong magnetic reflections appear at momentum transfer **Q** = $(Ha^*, Kb^*, q_{CDW})$ with $K =$ even, demonstrating direct coupling between incommensurate magnetism and the CDW in absence of the antiferromagnetic order parameter. Although we cannot provide a magnetic structure model based on the available data, the comparison of intensities at Miller indices $K = 0, 2$ suggests the presence of both $m_a$ and $m_b$ spin components (inset of Fig. 5e), consistent with longitudinal conical magnetic order.

## Discussion

As compared to transition metal dichalcogenides, where the magnetic ion is buried inside a rather symmetric block layer[34,35], $R$Te₃ harbors more complex structural features, with magnetic ions at the boundary between metallic and covalently bonded blocks. This mixed covalent/metallic environment for the magnetic ion is key to realizing the present scenario: it facilitates coupling between magnetic ions and a CDW on the tellurium square net, and – at the same time – generates unconventional magneto-crystalline anisotropy. In fact, the present charge-transfer phenomenology in the point charge model is partially

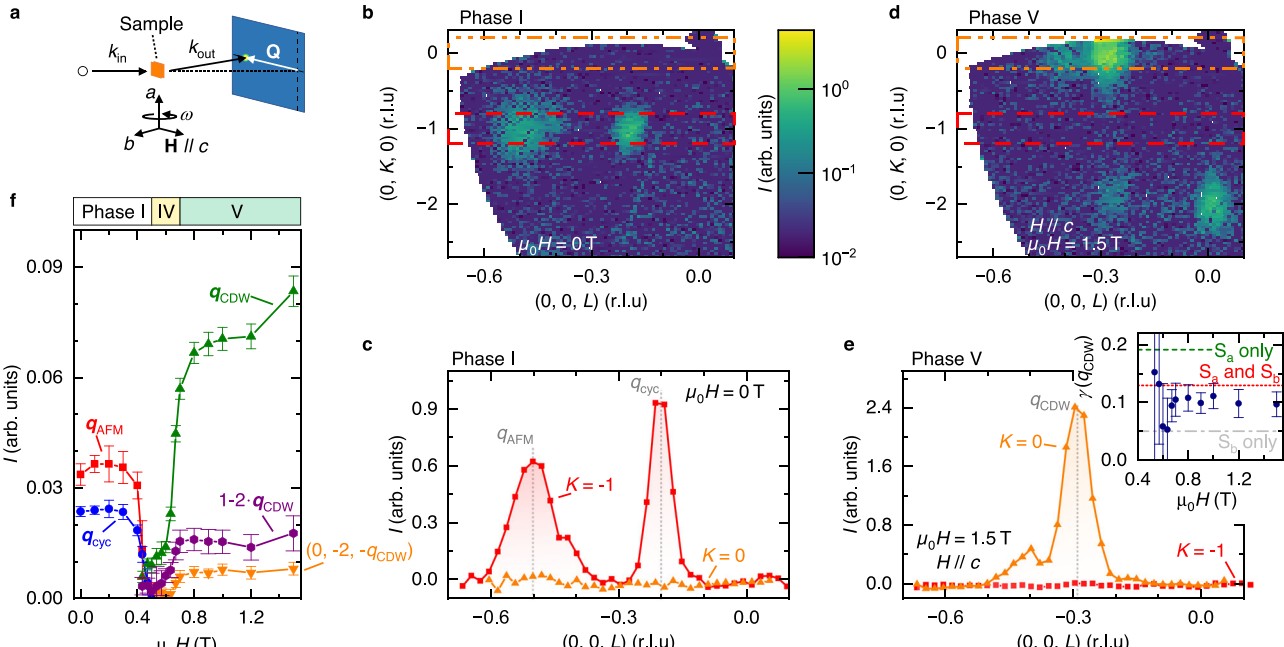

**Fig. 5 | Coupling of magnetic order and charge density wave in DyTe₃, tracked by small-angle neutron scattering (SANS) in a magnetic field along the *c*-axis at *T* = 2 K. a** Experimental geometry of the SANS experiment. Orange plate, blue rectangle, $k_{in}$, $k_{out}$, and **Q** are the sample, the area detector – shifted from the center axis to catch reflections with larger |**Q**| – incoming/outgoing neutron beam wavevector, and the momentum transfer of elastic scattering, respectively. A wedge of three-dimensional **Q** space is covered by rotating the sample in steps (angle *ω*, see "Methods"). **b**–**e** Scattering intensity in (0*KL*) cuts of momentum space (upper) and linecuts along (00*L*) (lower), for intensity integrated along the *K*-direction within the orange and red dashed boxes. The magnetic field induces a transition to ferroic stacking, *K* = 0. **f** Integrated intensity of various reflections as a function of magnetic field, covering three magnetic phases. Error bars correspond to statistical uncertainties of Gaussian fits to the extracted linecuts, c.f. **c**, **e**. Inset of **e**: experimental ratio $\gamma = I(0, -2, -0.293)/I(0, 0, -0.293)$ in Phase V, compared to simulations for fan-like orders (spin plane indicated, green and gray dashed lines) and a helimagnetic (longitudinal-cone, red dashed line) order. The experiments are consistent with a slightly distorted, conical state of the same period as the underlying charge order. Error bars are derived by error propagation from the statistical uncertainties in (**f**).

inspired by work on thin films of magnetic metals on insulating substrates[36], on electric field control of magneto-crystalline anisotropy[37], and on the behavior of magnetic materials when charge transfer is induced by oxidation at the surface[38].

Symmetry breaking with cycloid/spiral magnetic order of fixed helicity is rather widely observed in zigzag chain magnets (Supplementary Section IV A), but the present combination of antiferromagnetic and cycloidal components is unique. For example, $Mn_2GeO_4$ has cones arrayed on one-dimensional chains, with uniform cone direction along the chain[39]. In another metallic system, $EuIn_2As_2$, jumps in the rotation of a helimagnetic texture have recently been identified, with a short magnetic period[40]. In contrast, the rotation of moments in DyTe₃ proceeds with nearly parallel pairs of spins. We expect helimagnetic orders of the type observed here to be common in layered materials, and especially in rare earth tellurides and selenides that await exploration by neutron diffraction. Here, rich magnetic phase diagrams have been generally observed[41,42] and could be amenable to modeling by CDW-induced terms as in Eq. (1).

DyTe₃'s complex magnetic order, its relationship to a strain-controllable CDW[33], and its (likely) rich excitation spectrum certainly warrant further research. For example, the CDW's gapped collective excitation, termed Higgs mode, shows an axial character in $R$Te₃ as observed via Raman scattering experiments[15], and should thus have a spatially modulated (electric or magnetic) moment. The evolution of this unconventional CDW below $T_{N1}$ may provide insights on both the origin of magnetic order and the nature of the CDW in DyTe₃. Furthermore, the lowest energy, Goldstone mode of a typical helimagnet corresponds to a spatial shift of the magnetic texture, termed phason excitation[43]. In DyTe₃, the magnetic and CDW phasons[44] are expected to be closely intertwined, as evident from the robust $q_{cyc}(T)$ in Phase I,

and its jump – by the same amount as $q_{AFM}$ – in Phase II (Supplementary Fig. 13). Such locking between low-energy modes may have implications for dynamic responses, further enriching the spectrum of elementary excitations in $R$Te₃.

An important open question is the stability of helimagnetism in few-layer devices of DyTe₃. As a fundamental building block of the structure, we consider a DyTe slab sandwiched by Te square nets – that is half a unit cell in Fig. 1a. Being screened from top and bottom by tellurium layers, we expect no qualitative change of the local crystal field environment of Dy in the few-layer limit. However, the absence of an inversion center for odd numbers of layers, and its presence for even numbers of layers, may have a profound effect on magnetic ordering and the presence or absence of (right- or left-handed) helicity domains in the sample, considering the presence or absence of Dzyaloshinskii–Moriya interactions[45].

Most appealingly, DyTe₃ is a potential platform for spin-moiré engineering in solids, where complex magnetic textures can be designed by combining and twisting two or more helimagnetic sheets. Here, a plethora of noncoplanar spin textures can be engineered at will[6,46], while highly conducting tellurium square net channels may serve as a test bed for emergent electromagnetism in a tightly controlled setting[47,48].

## Methods

### Sample preparation and characterization

Single crystals are grown from tellurium self-flux following the recipe in ref. 42: we set elemental Dy and Te at a ratio of 1:21.65 in an alumina crucible, which in turn is sealed in a quartz tube in high vacuum. The raw materials are heated to 450 °C for 36 h and then to 780 °C in 96 h, where the melt remained for 48 h, followed by cooling to 450 °C at a

rate of 1.375 °C/h. The final product is centrifuged after renewed heating to 500 °C, so that plate-shaped single crystals of typical dimensions $5 \times 5 \times 1$ mm$^3$ are obtained. The face of each plate is perpendicular to the $b$-axis of DyTe$_3$'s orthorhombic unit cell, and facet edges tend to be parallel to either $a$ or $c$. The existence of impurity phases above 1% volume fraction is ruled out by single-crystal X-ray diffraction on cleaved surfaces in a Rigaku SmartLab X-ray powder diffractometer. The experiment yields lattice constants of $a = 4.27(2)$ Å, $b = 25.433(1)$ Å, and $c = 4.27(2)$ Å at room temperature, in good agreement with previous work[18]. We found it challenging to obtain high-quality powder X-ray data from crushed single crystals, which include traces of Te flux on their surface and form thin flakes, even when thoroughly ground in a mortar. We also verified the stoichiometric chemical composition of our crystals by energy-dispersive X-ray spectroscopy. Cleaved single crystals have a reddish-brown surface; but even in vacuum, the color of the surface changes to silvermetallic, and then to black, after 2 weeks or so. A red hue can be recovered by renewed surface cleaving.

## Magnetization measurements and crystal alignment

We use a commercial magnetometer with $T = 2$ K base temperature and a maximum magnetic field of 7 T (MPMS, Quantum Design, USA). The measurement is carried out using a rectangular-shaped single crystal of mass $m = 1.22$ mg, with carefully aligned edges along the $a$ and $c$ crystal axes. By means of a single-crystal diffractometer (Malvern Panalytical Empyrean, Netherlands), we confirm the extinction rule $h + k =$ even in space group $Cmcm$. It is difficult to distinguish $a$- and $c$-axes in this orthorhombic, yet nearly tetragonal structure by eye or with the help of a Laue diffractogram. Temperature dependent susceptibility $\chi(T)$ is measured in a DC magnetometer with an applied field of 1000 Oe; there is no observable difference between field-cooled and zero-field cooled magnetization traces. A demagnetization correction is carried out according to the standard expression $\mathbf{H}_{int} = \mathbf{H}_{ext} - N\mathbf{M}$, where $\mathbf{H}_{ext}$, $\mathbf{M}$, and $N$ are the externally applied magnetic field, the bulk magnetization, and the dimensionless demagnetization factor. The latter is calculated by approximating the crystal as an oblate ellipsoid[49]. We obtain the demagnetization-corrected susceptibility as $\chi_{int} = M / H_{int}$. For the $H - T$ phase diagram in Fig. 4a, the [101] direction is aligned within ±3° and bulk magnetization is measured in discrete field steps, for selected temperatures. Figure 4a shows data for decreasing magnetic field $\partial H/\partial t < 0$. Note that hysteresis occurs at all phase transitions shown in Fig. 4a, indicating their first-order nature. The magnetic anisotropy energy is expressed as $E/V_{uc} = K_0 + K_1 \cos^2(\theta) + \mathcal{O}[\sin^4(\theta)]$, where $\theta$ is the angle between $\mathbf{M}$ and the $b$-axis. Utilizing the free energy expression $F = F_0 + \sum_\alpha a_i(T)M_\alpha^2 + \mathcal{O}(\mathbf{M}^4)$ and the Curie–Weiss law $\chi_\alpha = C/(T - \Theta_{CW}^\alpha)$, where $\Theta_{CW}^\alpha$ is the Curie–Weiss temperature along the $\alpha \in (a, b, c)$ direction and $C = 2.077(1)$ K is the Curie constant of DyTe$_3$, we obtain $K_1 = (\mu_0/2)(M/V_{uc})^2(\Delta\Theta_{CW}/C) = 22(1)$ kJ m$^{-3}$, where $\Delta\Theta_{CW} = \Theta_{CW}^{ac} - \Theta_{CW}^b = -2.0(1)$ K is the difference between the Curie–Weiss temperatures in the $ac$-plane and along the $b$-axis (c.f. Fig. 1d, inset). Specific heat was recorded using a relaxation technique in a Quantum Design PPMS cryostat, in zero magnetic field. For specific heat anomalies in applied magnetic field, we employed the AC calorimetry technique in a custom-built setup. Anisotropy of the resistivity, as in Fig. 2b, was recorded on exfoliated flakes of thickness ~100 $\mu$m using the Montgomery technique. Electric contacts are made with Ag paste (Dupont) and deteriorate with time. To maintain excellent contact resistance ~1 Ω, it is crucial to immediately cool the contacted crystal in vacuum, after depositing the silver paste. The sample and contact quality is robust at low temperatures for at least 2 weeks.

## Elastic neutron scattering

We performed unpolarized and polarized neutron scattering experiments using the POlarized Neutron Triple-Axis spectrometer (PONTA)

installed at the 5G beam hole of the Japan Research Reactor 3. Two single crystals of DyTe$_3$ (Samples A and B) are cut into rectangular shapes with dimension $3.6 \times 2.7 \times 0.9$ mm and $1.7 \times 1.9 \times 0.8$ mm, respectively. For both samples, the widest surface is normal to the $b$-axis, and the sides are parallel to the $a$- or $c$-axis. Each sample is set in an aluminum cell, which is sealed with $^4$He gas for thermal exchange. We employed a $^4$He closed-cycle refrigerator with base temperature of 2.2 K, and measured intensities on the $(0, K, L)$ horizontal scattering plane. Using a PG (002) monochromator, the energy of the incoming neutron beam is set to $E_i = 14.7$ meV (30.5 meV) for upolarized measurements of Sample A (Sample B). For the unpolarized measurements, the spectrometer is operated in two-axis mode with horizontal beam collimation of open-80′-80′. In both unpolarized and polarized experiments, sapphire and pyrolytic graphite (PG) filters are installed between the monochromator and the sample, to suppress higher-order reflections from the monochromator to <0.5%. The observed integrated intensities are converted to structure factors after applying the Lorentz factor and absorption corrections.

For Sample B, we measured nuclear and magnetic Bragg reflections at 2.2 K by $\theta - 2\theta$ scans. For the scattering profiles showing a well-defined Gaussian-shape peak, we estimated the background from both ends of the profile. For the magnetic reflections located near the powder diffraction lines of the Al sample holder, we carried out background scans at 10 K, and subtracted the intensities from those measured at 2.2 K. We also measured the background data at 10 K for relatively weak commensurate magnetic reflections in the $Q$-range of $|Q| > 4.0$ Å$^{-1}$, to check for possible $\lambda/2$ contamination from the nuclear reflections. As for the absorption correction, we calculated the scattering path length $l$ inside the sample, based on the dimensions of Sample B and on the incident and scattered directions of the neutrons. The neutron transmission is given by $\exp(-\mu l)$, where $\mu$ is the linear absorption coefficient. Taking into account the incident energy, the absorption and the incoherent scattering cross-sections of DyTe$_3$, $\mu$ is calculated to be 8.392 cm$^{-1}$.

The diffraction profiles and integrated intensities shown in Figs. 2 and 3 were measured using Sample A. Contrary to integrated intensities in the case of refinement, the temperature dependences in Fig. 2c and Supplementary Fig. 13 are obtained from $L$-scans of magnetic scattering. The calculation of the scattering intensity in Fig. 2f, which includes the third harmonic reflection, takes into account instrumental resolution broadening (Supplementary Fig. 10), anharmonicity of the cycloidal magnetic structure component, and the presence of two magnetic domains (Supplementary Section IV).

Sample A is also used for polarized neutron scattering, in which the spectrometer is operated in the triple-axis mode with horizontal beam collimation of open-80′-80′-open. A polarized neutron beam with $E_i = 13.7$ meV is obtained by a Heusler (111) crystal monochromator. The spin direction of incident neutrons is set to be perpendicular to the scattering plane. We thus applied weak vertical magnetic fields of ~5 mT throughout the beam path by guide magnets and a Helmholtz coil. We used a Mezei-type $\pi$ spin flipper placed between monochromator and sample, and employed a Heusler (111) crystal analyzer to select the energy and spin states of scattered neutrons, separating SF and NSF intensities. The spin polarization of the incident neutron beam ($P_0$) is 0.823, as measured using the (002) nuclear Bragg reflection of the sample.

## Small-angle neutron scattering in magnetic field

SANS measurements were performed using the SANS-I instrument at Paul Scherrer Institute, Switzerland. A bulk single crystal of DyTe$_3$ (Sample E, $m = 73.4$ mg) was carefully aligned (c.f. crystal alignment in "Methods") and installed into a 1.8 T horizontal-field cryomagnet so that the $a$-axis is vertical, and the incident neutron beam is in the $bc$-plane. The magnetic field is applied parallel to the crystal $c$-axis, as shown in Fig. 5a. The incident neutron beam with $\lambda = 3.1$ Å wavelength (15% $\Delta\lambda/\lambda$) is

collimated over a distance of 4.5 m before the sample, and the scattered neutrons are detected by a 1 m² two-dimensional multidetector (pixel size 7.5 mm × 7.5 mm) placed 1.7 m behind the sample. To cover a broader $q$-space up to $q_{AFM}$ along the (01$L$) direction, the detector was also translated 0.45 m in the horizontal plane. For all SANS data, background signals from the sample and the instrument are subtracted using the data of the nonmagnetic state at $T = 10$ K and $\mu_0 H = 0$ T. The field-dependent SANS measurements are performed during a field-increasing process, after an initial zero-field cool to the base temperature of 2 K. For each measurement, rocking scans were performed, i.e., the cryomagnet is rotated together with the sample around the vertical crystal $a$-axes (rocking angle $\omega$) in a range from −102° to 55° and steps of 2° (−38° ≤ $\omega$ ≤ 28°) and 1° (else). Here $\omega = 0°$ is carefully aligned and corresponds to the configuration where the beam $\mathbf{k}_{in}$ is parallel to the crystallographic $b$-axis. The SANS maps shown in this paper are obtained by performing a 2D cut of the volume of reciprocal space measured through cumulative detector measurements taken at each angle of the rocking scan. For the SANS maps shown in Fig. 5, the integration width along the out-of-plane ($H$00) direction is ±0.15 reciprocal lattice units. The line cuts along (0$K_{fix}L$) shown in Fig. 5c, e are extracted by integrating over a region of ±0.2 reciprocal lattice units in the (0$K$0) direction. Peak positions and integrated intensities are calculated using those linecuts and a multi-peak fitting.

### Crystal electric field calculations
We use the software package PyCrystalField[50] for the calculation of CEF energies via the point charge model in the limit of strong spin–orbit interactions. The calculation is based on published fractional coordinates of Dy and Te ions within the crystallographic unit cell[18,22], with a ~0.15% tensile strain along the $a$-axis, lifting tetragonal symmetry and yielding finite $K_2$. In Supplementary Fig. 15, we vary the effective CEF originating from Te-A (on the Te₂ slab) by changing its point charge, while keeping the total charge in the environment of Dy unchanged. An unperturbed, diagonal Hamiltonian matrix is constructed from the energies in Supplementary Fig. 15, and the operator of total angular momentum $\mathbf{J} = \mathbf{L} + \mathbf{S}$ is also expressed in the basis of these CEF eigenstates. Adding an effective exchange field $E_{ex}J_\alpha$ ($\alpha = a, b, c$ are vector components), the total Hamiltonian is diagonalized and the expectation value of $J_a, J_b, J_c$ is evaluated in the respective ground state. The anisotropy constants are approximated, as

$$K_1 \propto \frac{1}{\langle J_b \rangle_b^2} - \frac{1}{\langle J_a \rangle_a^2} \tag{2}$$

$$K_2 \propto \frac{1}{\langle J_c \rangle_c^2} - \frac{1}{\langle J_a \rangle_a^2} \tag{3}$$

so that $K_1 < 0$ for easy-axis anisotropy along the $b$-axis and $K_2 > 0$ if $a$-axis orientation is energetically preferred over the $c$-axis. Here, $\langle J_a \rangle_\alpha$ is shorthand for $\langle \psi_{0,\alpha} | J_a | \psi_{0,\alpha} \rangle$, where $|\psi_{0,\alpha}\rangle$ is the ground state of the total Hamiltonian when an exchange field of magnitude $E_{ex}$ is applied along the $\alpha$-direction.

The charge density was calculated following the method in ref. [51]. Figure 4 shows iso-surfaces of the charge density at 75% of its maximum value. The anisotropic part of the charge density is exaggerated 20× in Fig. 4e according to the expression $20 \cdot (R - R_0) + R_0$, where $R_0$ corresponds to 10 Bohr radii. More details are given in Supplementary Figs. 15, 16.

### Spin model calculations
A model Hamiltonian, Eq. (1), is introduced from the viewpoint of symmetry in Supplementary Section I, to explain the essential experimental results. Here, magnetic frustration is lifted and the separation of spin components by modulation ($\mathbf{q}$-) vectors is explained naturally by the off-

diagonal $E_{CDW}^{ab}$ and $E_{CDW}^{ac}$ terms. An analytic solution is obtained in Fourier space, and variational calculations are carried out based on the spin ansatz and ignoring higher harmonics, for simplicity,

$$\begin{aligned}&\left( S_n^a, S_n^b, S_n^c \right) \\ &= \left( (-1)^n m, \sqrt{1 - m^2}\cos[(\pi + q_{CDW})z_n], \sqrt{1 - m^2}\sin[(\pi + q_{CDW})z_n] \right),\end{aligned} \tag{4}$$

In these terms, the energy is given by

$$E[m] = -J_2^{AFM} m^2 - \frac{E_{CDW}^{ab} + E_{CDW}^{ac}}{2} m\sqrt{1 - m^2} \tag{5}$$

and easily optimized at $m = m^*$ satisfying $\delta E[m^*] = 0$. The optimal antiferromagnetic moment $m^*$ gives the squared intensities $I_{AFM} = |S^a(q = \pi)|^2 = m^2$ and $I_{cyc} = |S^b(q = \pi + q_{CDW})|^2 + |S^c(q = \pi + q_{CDW})|^2 = (1 - m^2)/2$, depicted in Fig. 4c.

## Data availability
The data supporting the findings of this study are available from the authors upon reasonable request.

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

## Acknowledgements

We thank M. Nakano for permission to use his single-crystal X-ray diffractometer, and for support during the measurement. Moreover, we thank S. Kitou for initial advice on crystal field calculations, M. Kriener for support with experiments on magnetization and calorimetry, and Y. Kato for critical advice on the theoretical spin model. We thank P.M. Neves for support with the analysis of the wide-rocking angle SANS diffraction data. This work is based partly on experiments performed at the Swiss spallation neutron source SINQ, Paul Scherrer Institute, Villigen, Switzerland. This work was supported by JSPS KAKENHI Grant Nos. JP22H04463, JP22F22742, JP22K13998, JP23H01119, JP23KJ0557, and JP22K20348 as well as JST CREST Grant Number JPMJCR1874 and JPMJCR20T1 (Japan), and JST FOREST JPMJFR2238 (Japan). M.M.H. was funded by the Deutsche Forschungsgemeinschaft (DFG, German Research Foundation) – project number 518238332. The authors are grateful for support by the Fujimori Science and Technology Foundation, New Materials and Information Foundation, Murata Science Foundation, Mizuho Foundation for the Promotion of Sciences, Yamada Science Foundation, Hattori Hokokai Foundation, Iketani Science and Technology Foundation, Mazda Foundation, Casio Science Promotion Foundation, Inamori Foundation, Marubun Exchange Grant, and Kenjiro Takayanagi Foundation.

## Author contributions

Sh.A., M.H., and Y.O. synthesized and characterized the single crystals. S.E., M.H., and Sh.A. carried out calorimetry and magnetic measurements, while T.N., Se.A., Sh.A., Ri.Y., S.E., and M.H. carried out and analyzed elastic neutron scattering measurements, with extensive guidance from T.-h.A. S.E., J.S.W., and M.H. carried out and analyzed small-angle neutron scattering measurements. Electric transport measurements were carried out by S.E. and M.H., and theoretical modeling of crystal fields was conducted by S.E. under the guidance of T.-h.A. The magnetic structure was modeled by S.G., S.O., and Ry.Y.. The manuscript was written by M.H., M.M.H., and S.E. with contributions and comments from all co-authors.

## Competing interests

The authors declare no competing interests.
