## [Peer Review File · Nature Communications]

Reviewers' Comments:

Reviewer #1:

Remarks to the Author:

The authors report an extensive study of the specific heat, susceptibility, and neutron scattering of DyTe₃. Similar to other studies, their specific heat and susceptibility data reveal multiple magnetic transitions. Most interestingly, the careful neutron results present compelling evidence of a cycloidal magnetic order that appears to be tied to the CDW in these systems. The authors have carefully evaluated their data and compared it to theoretical calculations that strongly suggest the cycloid order arises from modulation of the exchange by the CDW and the unique crystal environment of the rare earth.

Given the growing interest in 2D multiferroics, CDW, and magnetic systems, I have no doubt this work will greatly interest the community. Furthermore, the paper is well-written, and the research is high quality. As such, I strongly recommend publication but ask the authors to consider some clarifications set out below.

A) The authors refer to the RTe₂ slabs as "insulating". This is probably correct but should be clarified why the authors believe this to be true (i.e. the fermi level is mostly derived from Te₂ bands). Especially given the transport in the c-axis is still quite metallic.

B) The authors claim they see a partial charge gap due to the magnetic order as the resistivity in the plane has increased anisotropy. However, this could also result from changes in spin-dependent scattering upon entering the magnetically ordered state.

C) On line 100-103 when initially discussing the Neutron data, it is not clear why the authors refer to the extra q scattering as cycloid (this becomes clear later). A sentence or two explaining this would be helpful to the reader. It might also be good to point out here the relationship to the CDW q and direction.

D) The authors refer to the recent work of the Burch group on the Axial Higgs in this system, but only as a study drawing "intense scrutiny" of the elementary excitations. In any event, that work shows the amplitude mode of the CDW has a moment, which would also form a natural basis for expecting the CDW and the magnetism to couple. This is another motivation to look carefully at the magnetism in RTe₃. Indeed, the authors' work demonstrates that such coupling occurs, thus suggesting further studies of changes in the Axial Higgs upon cooling into the magnetically ordered state. This is not commented on at all by the authors, but connecting their work to the recent results of Burch group does seem quite natural.

Reviewer #2:

Remarks to the Author:

The manuscript entitled "Non-coplanar helimagnetism in the layered van-der-Waals metal DyTe₃" by Dr. Esser and colleagues combines experimental and theoretical approaches to determine the magnetic properties, and in particular, the magnetic structure of the charge-density-wave (CDW) ordered van-der-Waals material DyTe₃. Heat capacity measurements reveal two clear phase transitions as a function of temperature, while a single anomaly in the magnetic susceptibility and electric resistivity suggests a magnetic origin and its effect on the electronic structure, respectively. Polarised neutron diffraction with longitudinal polarisation analysis is employed to deduce the constraints on the spin components of the magnetic structures within the two magnetically ordered phases which both reveal a commensurate and an incommensurate modulation. These constraints are then used to analyse the integrated intensities of magnetic Bragg peaks measured with unpolarised neutron diffraction in the ground state. The resulting conical magnetic structure, claimed to have a unique helicity despite the flip of the cone axis along the Dy zig-zag chains, is then corroborated by Monte Carlo simulations using a simplified J₁-J₂ model including the calculation of crystal electric field states for the 4f⁹ Dy electrons. The results are very well presented using aesthetically appealing figures and the presence of

helimagnetism in a layered van-der-Waals material with high electronic mobility would indeed make DyTe₃ a potential candidate for novel magnetic twistrionic or spintronic devices, if the magnetic structure is confirmed in the few-layer limit. Unfortunately, the central part of this manuscript - the experimental determination of the complex magnetic structure in a bulk sample - is based on single-crystal neutron diffraction data with insufficient quality, while their interpretation lacks the necessary thoroughness and is to some extent inconsistent or simply wrong, which I will detail here below.

1. The commensurate and incommensurate components of the ground state magnetic structure were analysed using 6 and 11 independent magnetic Bragg peaks which is a very limited data set bearing in mind that hundreds of reflections are routinely measured on single-crystal diffractometers in 4-circle geometry using samples with appropriate volume and/or magnetic moments of sufficient size, both being factors which are clearly fulfilled in this study. Especially the measurement of symmetry-equivalent reflections allows to minimise systematic errors in the data collection and increases its reliability due to the averaging of intensities and the reporting of the internal R value, a measure of the data quality. This is not the case in this study and no R_{int} can be reported, because no symmetry-equivalent reflections have been measured. Although some constraints on the spin components were deduced (yet incorrectly, see my next point) from polarised neutron scattering, the deduction of the complex cycloidal component requires a larger data set in order to be robust against different magnetic structure models. Even the commensurate part yields a rather bad agreement factor of $R = 0.146$ (note the discrepancy in the caption of Fig. E7b: $R = 0.12$), which should not be the case given the very simple antiferromagnetic spin alignment along the a axis and the treatment of purely magnetic reflections without any contribution of nuclear scattering. The agreement factor of the magnetic structure refinement would be expected to be comparable to the nuclear structure analysis which yields $R = 0.081$. Did the authors consider higher-order contaminations? They do not mention the use of a filter in the description of the diffraction experiments.

2. The polarised neutron diffraction experiments with longitudinal polarisation analysis (initial neutron beam polarised along the vertical axis) were carried out with the sample oriented in the (OKL) scattering plane, which - as the authors correctly state - allows to separate spin components m_a into non-spin-flip (NSF) scattering and components m_b and m_c into spin-flip (SF) scattering. The authors interpret the comparable SF intensities of the $(0\ 1\ 1+q_{cyc})$ and $(0\ 9\ q_{cyc})$ reflections (Fig. 3e,f) - being almost orthogonal to each other - as a sign for comparable m_b and m_c components (line 122-124). This is simply wrong as they do not take into account the amplitude of the magnetic interaction vector M_{perp} (the component of the magnetic structure factor M perpendicular to the scattering vector Q). As an example: Does the absence of magnetic scattering on the $(0\ 10\ q_{cyc})$ reflection prove the zero m_c coefficient? Of course not, because the $(0\ 10\ q_{cyc})$ reflection is forbidden ($M = 0$) by the magnetic symmetry. It is therefore not correct to deduce equal m_b and m_c components from two comparable intensities without taking into account the respective structure factors. Drawing this wrong conclusion and calculating the theoretical intensities (M_{perp}^2 times Lorentz factor) leads to very different results as shown in my next point. Another false interpretation of polarised neutron data is apparent in the caption of Fig. E7, where the authors interpret the $P/P_0 = -1$ values for magnetic Bragg reflections at different omega angles as a sign for m_b being of similar amplitude as m_c . An observed value of $P/P_0 = -1$ for different magnetic reflections at different omega angles simply states the absence of NSF scattering and in consequence that the magnetic moments lie in the b - c plane (taking into account the experimental geometry), but it cannot be concluded that m_b is similar to m_c . In fact, any moment distribution in the b - c plane would yield $P/P_0 = -1$ for all (OKL) reflections with finite intensity.

3. The integrated intensities have been analysed by including two magnetic domains. Apart from the spin components a phase angle δ was refined for the cycloidal part, which represents the phase shift between the upper and lower cycloids of a zig-zag chain. Using the nomenclature of Dy_j in Fig. 3g, Dy₂ is displaced by $z = 1/2$ from Dy₁ and therefore the angle between those two spins, for $\delta = 0$, would be $q_{cyc} * z = 0.21 * 1/2$ in fractions of 2π , i.e. 37.8° . A reported phase angle of

58° would therefore mean an angle between the Dy₁ and Dy₂ spins of 37.8° + 58° = 95.8° which corresponds well to Fig. 3g and Fig. E7e (on the other hand, in line 556 a phase angle of $\delta \sim 2\pi/5 = 72$ degrees is reported, is 58° ~ 72°??, 58° is rather $2\pi/6$). In line 549 it is mentioned that a single parameter +/- δ is refined for both domains, even though in the caption of Fig. E7e it is described that the phase angle was refined individually in both domains. There is a lack of consistence here. I will assume that δ has the same value but different sign in the two domains, but then the domain with $-\delta$ does not agree with the magnetic structure shown in Fig. E7d. The spin of Dy₂ seems to have an angle of -58° with Dy₁, instead of 37.8° - 58° = -20.2°. This is again inconsistent. Assuming that just the picture is wrong, I reproduced the magnetic structures in both domains (using FullProf and Mag2Pol, two freely available programs to analyse diffraction data) and obtain that the intensity of the (0 1 1+q_{CYC}) reflection is about 2 times stronger than that of the (0 9 q_{CYC}) reflection (including the Lorentz factor). If I use the presumably wrongly depicted magnetic structure in Fig. E7d (which does not have any symmetry relation to the first domain), the (0 1 1+q_{CYC}) reflection would be roughly 50% stronger in intensity than the (0 9 q_{CYC}) reflection. This does not agree with the raw data shown in Fig. 3e,f (polarised neutrons) and shows the consequence of its misinterpretation (see my previous point). Yet, the model apparently explains the unpolarised neutron data very well (Fig. E7c), which constitutes a serious problem of consistency and sheds doubts on the validity of the results. In the same line, why didn't the authors comment on the different intensities of the commensurate reflections shown in Fig. 3a,b?

4. As pointed out in my previous point, there may be some errors in the authors' calculations of structure factors or the data are unreliable and compatible with different magnetic structure models. Did the authors use their proper code for the calculations and refinements? They do not mention any software package known in the neutron diffraction community. But apart from that, there is another major problem with the data analysis. The phase angle δ is not a refinable parameter as the Dy ions in the 4 different sheets (labelled j = 1 to 4) are related by symmetry. The little group of space group C₂cm with a propagation vector (0 1 0.21) contains 4 symmetry operators in the (000)+ set, which are the identity, a 2-fold screw axis in 0,0,z, a glide plane c in x,0,z and a mirror plane in 0,y,z. These 4 symmetry operators together with the coset generated by the C-centering generate the 4 Dy positions in the conventional unit cell. The atoms Dy₂ and Dy₄ are related to Dy₁ and Dy₃ by both the 2-fold screw axis and the glide plane c (due to the special position 4c with x = 0), which invert the rotation sense of the cycloid. The cycloids on sheets j=2 and j=4 rotate therefore in the opposite sense in comparison to those on j=1 and j=3. In consequence, the magnetic structure - respecting the underlying nuclear symmetry - cannot induce a polarisation along the b axis as claimed in the caption of Fig. 1c and does not possess a fixed sense of rotation as indicated in line 65 ^{SEP}.

The derivation of complex magnetic structures requires an in-depth symmetry analysis. The two domains mentioned by the authors result from a loss of a symmetry operator at the transition into the magnetically ordered state. The description of the magnetic symmetry - at least using irreducible representations, but ideally using magnetic space groups and superspace groups, is mandatory. In fact, the 2 domains of the q_{AFM} structure can be described by the two basis vectors of a two-dimensional irreducible representation or alternatively by the two magnetic space groups C₂/m (-b+c, a, c; 0, 0, 1/4) and C₂/m (b+c, -a, c; 0, 0, 1/2) (note the different basis transformations for the two domains). The q_{CYC} structure follows a one-dimensional irreducible representation or a magnetic superspace group, which yields of course a different spin configuration than the one presented by the authors. The authors have not considered group theory aspects and present a solution which cannot be justified based on the data at hand, i.e. without further proof of symmetry reduction.

5. The attribution of J₁ and J₂ to the nearest-neighbour (NN) and next-nearest-neighbour (NNN) interactions is in principle very simple, but there are several inconsistencies throughout the manuscript. In line 58 it is mentioned that the NN is located in the respective other layer. While this seems to match with the perspective sketches of the zig-zag chains, it is not what Fig. 1a,b suggest and not what can be derived based on the atomic parameters of either the average C₂cm structure or those of the superspace group C₂cm(00g)000 from Ref. 18. The distance corresponding to J₁ (equivalent to the c lattice parameter) is shorter than that corresponding to

J_2 (c.f. Fig. 1b). In this respect, the authors mention that they have used the structural parameters from Ref. 18 to calculate the structure factors (line 524), but they do not mention the superspace group formalism or the determined superspace group. The atomic positions in Ref. 18 are in fact incompatible with space group Cmcm (e.g. Dy at position 0.94 0.169803(12) 0.25, note the non-zero x component), and therefore the citation of Ref. 51 (Squires, line 526) is inappropriate and inconsistent as it does not treat the superspace group formalism. Also, the authors should have discussed how the structural superspace group C2cm(00g)000 - being polar along the a axis - can be in agreement with a supposedly polar distortion along the b axis (see caption of Fig. 1c) via the reported magnetic structure. The manuscript lacks a decisive point here, as no symmetry analysis was carried out.

6. The previous point is also an issue with the authors' J_1 - J_2 model used in their Monte Carlo simulations. In Fig. 1b the NN interaction J_1 is defined between two Dy ions on the same sheet (i.e. same atomic position y), while the NNN interaction J_2 is defined between two sheets with different y -values. In Fig. E2 the authors claim that the in-phase model is favoured by the modulated $J_{1,CDW}$, but refer to spins of different layers to compute the dot product. This is not consistent with the model and the definition of the exchange constants. And I have a further problem of understanding: if J_1 and J_2 are modulated (as defined in lines 162-163) by $q_{CDW} = 0.29$ (note the different value of 0.3 in line 589 and the mistake in line 164, $0.5 - 0.29 = 0.31$, as well as the use of x and z in the sine term, line 163 and 588, respectively), all values between $+J_{1,CDW}$ and $-J_{1,CDW}$ will be adopted for J_1 along a long enough chain along the z direction. The argument that the in-phase model is favoured by J_1 is therefore insufficient since large positive amplitudes (depicted by the grey shaded bars in Fig. E2) will be multiplied with positive and negative J_1 values and therefore cancel out in the energy balance. If the authors meant $J_1 = J_1(0) + J_{1,CDW} \sin(q_{CDW} * z)$ (note that the $J_1(0)$ term is not mentioned in the manuscript), then the interval of J_1 values would be $[J_1(0) - J_{1,CDW}, J_1(0) + J_{1,CDW}]$, but there would still be a problem in my opinion: It has to be noted that the compensation between positive and negative amplitudes of $m_i * m_{i+1}$ (e.g. $m_1 * m_2$ and $m_2 * m_3$ in Fig. E2a) is the same between the two models depicted in Fig. E2. Using the reported values of 6.49 μ_B for the AFM component and 6.18 μ_B for the cycloidal component, the angles between different spin pairs in the conical magnetic structure can be calculated for the in-phase and the out-of-phase models. In the in-phase model, the large positive amplitude originates from nearly parallel spins with angle 13.86° [$\cos(13.86^\circ) = 0.971$], while the negative amplitude comes from spins with an angle of 124.9° [$\cos(124.9^\circ) = -0.572$]. The values for the out-of-phase model are 61.53° [$\cos(61.53^\circ) = 0.477$] and 94.48° [$\cos(94.48^\circ) = -0.078$], respectively. In both cases the sum of the amplitudes is 0.399. It is therefore not clear, why one model would be preferred over the other by J_1 since the energy balance in the Hamiltonian would be the same.

7. In lines 537 and 550 the authors mention that they have assumed equal domain populations for both the q_{AFM} and q_{CYC} components. While this is of course a reasonable assumption, they should have refined the populations against their unpolarised neutron data in order to demonstrate the robustness of their model. If the cycloidal and the antiferromagnetic components are indeed coupled to an in-phase structure then the domain population should be coupled as well, i.e. similar domain ratios should result for the different magnetic structure components. However, refining the two main axes of the cycloidal envelope (m_b , m_c), an eventual phase angle δ and a domain population approaches or exceeds the reasonable number of parameters for a data set of 11 measured magnetic Bragg peaks.

8. Fig. E8 shows the temperature dependence of line scans in reciprocal space both using polarised and unpolarised neutrons. Fig. E8a (unpolarised) shows how the q_{CYC} peak position shifts from phase I to phase II, and there is a coexistence of both peaks at $T = 3.7$ K, while only one peak is observable at $T = 3.8$ K. Why is there a signature of both peaks in the SF scattering data shown in Fig. E8b at 3.8 K? Furthermore, the authors do not present supporting data and analysis for the magnetic structure in phase II depicted in Fig. E8d. The polarised neutron line scans in Fig. E8b,c indicate the reduction of the m_c component, but without a refinement to integrated intensity data

it is impossible to deduce the phase angle δ which the authors apparently consider to be constant throughout the phase transition. There is no reason to assume this, and there is little information about how this structure was deduced.

Minor details, but making the review more difficult, are the following inconsistencies in figure referencing:

Caption Fig. 2e: There is no panel h.

Caption Fig. 2f: Panel d does not show a line scan.

Beginning of caption Fig. 3: Fig. 2g does not exist

Line 145-146: Figs. E6 and E7 refer to the ground state and not to phase II

The caption of Fig. 4d is not sufficiently descriptive

Line 84: Fig. 2c is not a thermodynamic or transport probe

Given the limited neutron data quality, the numerous inconsistencies in the neutron data analysis, to some extent wrong conclusions and the absolute lack of symmetry considerations, it is difficult to trust the validity of the presented results concerning the ground state magnetic structure in DyTe₃, which is the key result of this study. Under these circumstances I am sceptical that even a heavily revised version of the manuscript can be considered for publication in Nature Communications.

Response letter 'Noncoplanar helimagnetism in the van-der-Waals magnet DyTe₃'

(S. Akatsuka *et al.*)

We thank the Editor and the two Reviewers for extensive comments and careful reading of our manuscript. Reviewer #1 emphasizes that "the paper is well-written, and the research is high quality", while recommending publication given certain changes. Reviewer #2 criticizes the analysis and completeness of the neutron scattering data. We have aimed to address these concerns by taking an entirely new diffraction data set on a second sample, while also answering in detail to his / her comments about symmetry analysis and parts of the data analysis. Reviewer #2 also acknowledges the general interest of our work, and the suitable presentation of the data and text (while providing many useful edits). We appreciate that both Referees have sent us thorough and sincere remarks, which have helped us improve this manuscript.

Below, we have spent a lot of effort to lay out our response to all the Reviewers' questions in a point-by-point manner, highlighting the Referees' text by bold font for clarity.

Response to Reviewer #1:

The authors report an extensive study of the specific heat, susceptibility, and neutron scattering of DyTe₃. Similar to other studies, their specific heat and susceptibility data reveal multiple magnetic transitions. Most interestingly, the careful neutron results present compelling evidence of a cycloidal magnetic order that appears to be tied to the CDW in these systems. The authors have carefully evaluated their data and compared it to theoretical calculations that strongly suggest the cycloid order arises from modulation of the exchange by the CDW and the unique crystal environment of the rare earth.

Given the growing interest in 2D multiferroics, CDW, and magnetic systems, I have no doubt this work will greatly interest the community. Furthermore, the paper is well-written, and the research is high quality. As such, I strongly recommend publication but ask the authors to consider some clarifications set out below.

We thank the Referee for a careful reading of our work, and for acknowledging its merit. We have addressed all the Referee's questions in hope of further improving this manuscript.

A) The authors refer to the RTe₂ slabs as "insulating". This is probably correct but should be clarified why the authors believe this to be true (i.e. the fermi level is mostly derived from Te₂ bands). Especially given the transport in the c-axis is still quite metallic.

The Referee is correct that we oversimplified the discussion of conducting properties, originally aiming to emphasize the more covalent nature of Te-B bonds as compared to Te-A bonds (Fig. 4). We have removed this language from the main text, and replaced "insulating" with "covalently bonded" everywhere in the text and SI. We believe this language is reasonable, as many previous electronic structure calculations show that the

Fermi surface in $R\text{Te}_3$ can be described by quasi-two dimensional sheets: e.g. V. Brouet *et al.*, Phys. Rev. B **77**, 235104 (2008). We also added a paragraph to the SI:

“Anisotropic electronic transport properties:

Angle-resolved photoemission (ARPES) studies combined with a tight binding (TB) model in Ref. [Brouet *et al.*, 2008] reveals that, in the family of the rear-earth tritellurides $R\text{Te}_3$, the states at the Fermi level are mainly formed by the in-plane p_x and p_z orbitals of the Te-A ions in the Te square net, c.f. Fig. 1a. Indeed, they are well separated by more than 1 eV from other bands, already indicating anisotropic bonding and transport properties. With a standard four-probe method and a modified Montgomery geometry, Ru *et al.* determined both in-plane (ρ_{ac}) and out-of-plane resistance (ρ_b) of various $R\text{Te}_3$ compounds. They differ by at least one order of magnitude ($\rho_b \gg \rho_{ac}$), consistent with a metallic Te square net as well as covalently bonded $R\text{Te}$ slabs.”

B) The authors claim they see a partial charge gap due to the magnetic order as the resistivity in the plane has increased anisotropy. However, this could also result from changes in spin-dependent scattering upon entering the magnetically ordered state.

The point is well acknowledged. For example, in many magnetic materials, the resistivity drops at T_C or T_N due to the suppression of magnetic fluctuations in the ordered state. In this scenario, strong scattering is present at high temperatures due to fluctuations, which is suppressed abruptly upon cooling.

We have added a figure “Anisotropy of electrical transport properties in ac basal plane of DyTe_3 ” to the Extended Data, showing the anisotropic resistance of two single crystals over a larger temperature range.

A large anisotropy appears in the single- \mathbf{Q} CDW state below 320 K, but is suppressed at lower temperature in the previously reported double- \mathbf{Q} CDW order ($T < 30$ K), where a plateau forms in ρ_c / ρ_a . Only just around the magnetic phase transition T_N , we observe a further sharp jump of the anisotropy. If strong (unidirectional) scattering did impact ρ_c / ρ_a , we may expect significant deviation from the plateau value already at higher temperatures (say, $T \sim 12$ K). The absence of such, and the sharpness of the decrease at T_N , are evidence (to us) of the gap opening scenario below T_N .

We have further added a comment to the SI, regarding a second point in favor of the gap opening scenario:

“There is some ambiguity in the interpretation of resistance anisotropy changes at T_N , which can be ascribed either to opening of a partial charge gap in the ordered state, or to fluctuations in the paramagnetic regime and their suppression below T_N . In DyTe_3 , the sign of the observed change implies that, when moving from the paramagnetic into the ordered regime, the resistance along the c -axis becomes larger than the resistance along the a -axis. This result can be neatly explained by attributing partial gap opening to $\mathbf{q}_{\text{AFM}} = (0, b^, 0.5c^*)$, $\mathbf{q}_{\text{cyc}} = (0, b^*, 0.207c^*)$, while unidirectional fluctuations should naively enhance ρ_c / ρ_a above T_N , and suppress it below T_N .”*

C) On line 100-103 when initially discussing the Neutron data, it is not clear why the authors refer to the extra q scattering as cycloid (this becomes clear later). A sentence or two explaining this would be helpful to the reader. It might also be good to point out here the relationship to the CDW q and direction.

We thank the Referee for pointing out this weakness in the manuscript. New text was inserted into the main text as follows:

“A commensurate (C) reflection $\mathbf{q}_{AFM} = (0, b^*, q_{AFM})$, $q_{AFM} = 0.5 c^*$; an incommensurate (IC) reflection $q_{cyc} = (0, b^*, q_{cyc})$, $q_{cyc} = 0.207 c^*$, where $b^* = 2\pi/b$ and $c^* = 2\pi/c$ are reciprocal lattice constants (Methods). There is also a higher harmonic (3Q) reflection, corresponding to three times the length of q_{cyc} , which describes an anharmonic distortion of the texture. As a main result of this work, we ascribe q_{cyc} to a cycloidal structure in the magnetic ground state of DyTe₃, that results from a coupling $\mathbf{q}_{AFM} \pm \mathbf{q}_{CDW}$ between the C order and a charge-density wave (CDW) modulation \mathbf{q}_{CDW} at $(0, 0, q_{CDW})$, $q_{CDW} = 0.29 c^*$ in the rare earth tritelluride family.”

D) The authors refer to the recent work of the Burch group on the Axial Higgs in this system, but only as a study drawing "intense scrutiny" of the elementary excitations. In any event, that work shows the amplitude mode of the CDW has a moment, which would also form a natural basis for expecting the CDW and the magnetism to couple. This is another motivation to look carefully at the magnetism in RTe₃. Indeed, the authors' work demonstrates that such coupling occurs, thus suggesting further studies of changes in the Axial Higgs upon cooling into the magnetically ordered state. This is not commented on at all by the authors, but connecting their work to the recent results of Burch group does seem quite natural.

We thank the Referee for raising the point of coupled CDW and magnetic properties in DyTe₃, which were insufficiently addressed in the previous version of the manuscript.

Firstly, to bring out more clearly the direct coupling of charge and spin order, we have conducted Small Angle Neutron Scattering (SANS) experiments in a magnetic field along the c -axis (parallel to \mathbf{q}_{CDW}), with the aim of “simplifying the magnetic structure”. Indeed, we observe a transition from two magnetic modulation vectors (\mathbf{q}_{AFM} , $\mathbf{q}_{AFM} \pm \mathbf{q}_{CDW}$) to a single magnetic modulation \mathbf{q}_{CDW} (Fig. 5 in the revised manuscript). This is also reminiscent of recent observations in GdTe₃, using scanning tunneling microscopy (arXiv:2308.15691).

Second, based on these observations and on our improved understanding of magnetic interactions in this material, we have revised the spin Hamiltonian, now a combination of Ising exchange (along the a -axis) and anisotropic exchange in the basal plane, which is facilitated by CDW order. A more detailed discussion is provided in our response to Referee 2 and in section “Spin Hamiltonian” of the revised Extended Data.

Thirdly, we have revised the Conclusion section of the main text to further discuss the impact of Wang *et al.*'s work on the present findings:

“For example, the CDW's gapped collective excitation, termed Higgs mode, shows a magnetic character in $R\text{Te}_3$, as observed via Raman scattering experiments, and its evolution below T_{N1} may provide insights on both the origin of magnetic order and the nature of the CDW state in DyTe_3 .”

In summary of all questions and comments, we would like to thank the Referee again for helpful corrections and suggestions.

Response to Reviewer #2

The manuscript entitled "Non-coplanar helimagnetism in the layered van-der-Waals metal DyTe_3 " by Dr. Esser and colleagues combines experimental and theoretical approaches to determine the magnetic properties, and in particular, the magnetic structure of the charge-density-wave (CDW) ordered van-der-Waals material DyTe_3 . Heat capacity measurements reveal two clear phase transitions as a function of temperature, while a single anomaly in the magnetic susceptibility and electric resistivity suggests a magnetic origin and its effect on the electronic structure, respectively. Polarised neutron diffraction with longitudinal polarisation analysis is employed to deduce the constraints on the spin components of the magnetic structures within the two magnetically ordered phases which both reveal a commensurate and an incommensurate modulation. These constraints are then used to analyse the integrated intensities of magnetic Bragg peaks measured with unpolarised neutron diffraction in the ground state. The resulting conical magnetic structure, claimed to have a unique helicity despite the flip of the cone axis along the Dy zig-zag chains, is then corroborated by Monte Carlo simulations using a simplified J_1 - J_2 model including the calculation of crystal electric field states for the $4f^9$ Dy electrons.

We thank the Reviewer for spending her / his time to study our manuscript, and for providing many comments that are addressed in the point-by-point response below, including a large body of new neutron data and discussion.

We apologize for the length of the text and the length of the revised manuscript, but the comments from the Referee were extensive. The manuscript has certainly benefited from the Referee's comments, especially as regards the symmetry analysis.

The results are very well presented using aesthetically appealing figures and the presence of helimagnetism in a layered van-der-Waals material with high electronic mobility would indeed make DyTe_3 a potential candidate for novel magnetic twistrionic or spintronic devices, if the magnetic structure is confirmed in the few-layer limit.

We are thankful to the Referee for stressing the strong points of our work, while also rising points of criticism that we have aimed to address in the answers below.

Unfortunately, the central part of this manuscript - the experimental determination of the complex magnetic structure in a bulk sample - is based on single-crystal neutron diffraction data with insufficient quality, while their interpretation lacks the necessary thoroughness and is to some extent inconsistent or simply wrong, which I will detail here below.

We acknowledge the Referee's sincere concerns about the data presentation and analysis in our prior manuscript. After additional experiments, we spent several months to respond to the Referee's comments, including much discussion in the more systematic framework of magnetic space groups, as shown below.

1. The commensurate and incommensurate components of the ground state magnetic structure were analysed using 6 and 11 independent magnetic Bragg peaks which is a very limited data set bearing in mind that hundreds of reflections are routinely measured on single-crystal diffractometers in 4-circle geometry using samples with appropriate volume and/or magnetic moments of sufficient size, both being factors which are clearly fulfilled in this study. Especially the measurement of symmetry-equivalent reflections allows to minimise systematic errors in the data collection and increases its reliability due to the averaging of intensities and the reporting of the internal R value, a measure of the data quality.

We thank the Referee for comments on the dataset and recommended analysis. To address the Referee's concern, we have collected an entirely new, second dataset with a second single-crystalline sample (sample B) from a second batch of DyTe₃, in October 2023. We again used beamline PONTA-5G of JRR-3 for this task, increased the neutron energy, and were able to access a total of 108 magnetic reflections (all independent) in the ground state of DyTe₃.

There are many changes and modifications to the text, for example in Methods, where we discuss the improved analysis of the new dataset including correction for the absorption effect and subtraction of powder line backgrounds. We also added sections "Symmetry and Structure Factor: Commensurate Component in Phase I" and "Symmetry and Structure Factor: Incommensurate Component in Phase I" to the Extended Data. In the interest of brevity of this report, we keep the discussion short, but we show the refinement of the new and old data, side by side, in Fig. R1.

Figure R1. Magnetic structure refinement for the ground state of DyTe₃. Antiferromagnetic (commensurate, \mathbf{q}_{AFM}) and cycloidal (incommensurate \mathbf{q}_{cyc}) components are refined separately from neutron scattering data, without polarization analyzer, for sample A and B, respectively. **a, c**, Averaging the two antiferromagnetic domains $uudd$ and $uddu$ with equal weight, we find good agreement between the measured structure factor F_{obs} and model calculations F_{cal} . The reliability factors are given in the respective figure panels. **b, d**, Likewise, good agreement is found when summing two δ -domains for the incommensurate component \mathbf{q}_{cyc} . Here, the value of the phase shift δ between sheets in a bilayer, as defined in Fig. 3g, was optimized to minimize R . For the refinement of \mathbf{q}_{cyc} of sample A, undistorted ($m_b = m_c$) cycloidal order was assumed, but relaxing this condition yields only marginal improvement of R at $m_c / m_b = 1.03$. For sample B, the elliptic distortion was a free parameter in the model (section “Results of Magnetic structure analysis” in Extended Data).

As compared to the previous data set on sample A, the new data set was improved by (a) choosing a smaller, more isotropic crystal and longer integration times, hence suppressing \mathbf{Q} -dependent absorption, (b) increasing the neutron energy and accessing a larger region of \mathbf{Q} -space, (c) using θ - 2θ for all reflections, (d) measuring a high temperature background, which is important especially at large 2θ values where the magnetic intensities become weak, (e) quantitative absorption correction for all reflections. Although we show the data side by side in Fig. R1, we have updated the manuscript by replacing the refinement data of sample A by new refinement data from sample B. Below, we further discuss the limitations of absolute intensities obtained from sample A.

We also would like to note that, given the information provided by polarized neutron scattering, we believe that a – admittedly more comprehensive – four-circle experiment would not qualitatively change the results of this analysis, or provide information far beyond what is achievable using the present, high-symmetry scattering plane.

This is not the case in this study and no R_{int} can be reported, because no symmetry-equivalent reflections have been measured.

We thank the Referee for recommending the use of R_{int} , which is used by some researchers in this field, especially when the number of reflections is very large and when the sample is isotropic in shape. In reality, symmetry-equivalent reflections can have different intensities due to the absorption and extinction effects, both of which depend on the scattering path length inside of the sample; the path lengths are dependent on the shape of the sample, scattering geometry, azimuthal angle, and so on. In the present case, there are constraints about crystal growth and preparation – DyTe_3 grows in the shape of thin plates, and it is not possible to polish the sample into the shape of a sphere. Even cutting the crystal by a razor blade is risky due to potential peak broadening / introduction of stacking faults.

For the revised data set on sample B, we have added a careful absorption correction based on the thin plate shape of the sample. The revised Methods section specifies the details of our measurement on sample B as follows:

“For sample B, we measured nuclear and magnetic Bragg reflections at 2.2 K by θ - 2θ scans. For the scattering profiles showing a well-defined Gaussian-shape peak, we estimated the background from both ends of the profile. For the magnetic reflections located near the powder diffraction lines of the Al sample holder, we carried out background scans at 10 K, and subtracted the intensities from those measured at 2.2 K. We also measured the background data at 10 K for relatively weak commensurate magnetic reflections in the \mathbf{Q} -range of $|\mathbf{Q}| > 4.0 \text{ \AA}^{-1}$, to check for possible $\lambda / 2$ contamination from the nuclear reflections. As for the absorption correction, we calculated the scattering path length l inside the sample, based on the dimensions of Sample B and on the incident and scattered directions of the neutrons. The neutron transmission is given by $\sim \exp(-\mu l)$, where μ is the linear absorption coefficient. Taking into account the incident energy and the absorption and incoherent scattering cross-sections of DyTe_3 , μ is calculated to be 8.392 cm^{-1} .”

Although some constraints on the spin components were deduced (yet incorrectly, see my next point) from polarised neutron scattering, the deduction of the complex cycloidal component requires a larger data set in order to be robust against different magnetic structure models. Even the commensurate part yields a rather bad agreement factor of $R = 0.146$ (note the discrepancy in the caption of Fig. E7b: $R = 0.12$), which should not be the case given the very simple antiferromagnetic spin alignment along the a axis and the treatment of purely magnetic reflections without any contribution of nuclear scattering. The agreement factor of the magnetic structure refinement would be expected to be comparable to the nuclear structure analysis which yields $R = 0.081$. Did the authors consider

higher-order contaminations? They do not mention the use of a filter in the description of the diffraction experiments.

We thank the Referee for pointing out limitations of the neutron data analysis. We used a pyrolytic graphite (PG) filter to suppress $\lambda/2$ scattering in the experiments for both sample A and B; the amount of contamination is roughly 0.5% of the primary peak intensity that is seeping into the secondary peak. For sample B, we were more careful to subtract backgrounds from powder lines, but in the measurement of sample A we generally avoided \mathbf{Q} -positions close to powder lines. We also carried out absorption correction for sample B. Regarding the quality of fit as characterized by R , a problem in the previous data set (sample A) was the limited number of reflections, as the Referee points out. However, we note the typical criterion of $R < 0.15$ for reasonable agreement of data and model was satisfied for all our data, even in the previous submission.

In the revised dataset for sample B, c.f. Fig. R1, the R -value of the magnetic refinement for the AFM and incommensurate components is improved to $R < 0.1$, comparable in magnitude to the R -value of the structural refinement for the same sample. Increasing the degrees of freedom in the fit to the magnetic data, such as the ratio of domains, we may further improve the R -value (see discussion below).

2. The polarised neutron diffraction experiments with longitudinal polarisation analysis (initial neutron beam polarised along the vertical axis) were carried out with the sample oriented in the (OKL) scattering plane, which - as the authors correctly state - allows to separate spin components m_a into non-spin-flip (NSF) scattering and components m_b and m_c into spin-flip (SF) scattering. The authors interpret the comparable SF intensities of the $(0\ 1\ 1+q_{cyc})$ and $(0\ 9\ q_{cyc})$ reflections (Fig. 3e,f) - being almost orthogonal to each other - as a sign for comparable m_b and m_c components (line 122-124). This is simply wrong as they do not take into account the amplitude of the magnetic interaction vector M_{perp} (the component of the magnetic structure factor M perpendicular to the scattering vector \mathbf{Q}). As an example: Does the absence of magnetic scattering on the $(0\ 10\ q_{cyc})$ reflection prove the zero m_c coefficient? Of course not, because the $(0\ 10\ q_{cyc})$ reflection is forbidden ($M = 0$) by the magnetic symmetry. It is therefore not correct to deduce equal m_b and m_c components from two comparable intensities without taking into account the respective structure factors. Drawing this wrong conclusion and calculating the theoretical intensities (M_{perp}^2 times Lorentz factor) leads to very different results as shown in my next point.

We thank the Referee for this comment, and for pointing out incorrect phrasing in the manuscript. We have corrected the language in the caption of Fig. 3 and removed references to “equal moment length”:

“g, Derived magnetic structure model for q_{cyc} , where m_a (NSF) vanishes while m_c and m_b (SF) both appear at comparable magnitudes in phase I.”

In our response to question 3 below, we further consider the relative intensity of various reflections in Fig. 3 as suggested by Referee 2.

Another false interpretation of polarised neutron data is apparent in the caption of Fig. E7, where the authors interpret the $P/P_0 = -1$ values for magnetic Bragg reflections at different omega angles as a sign for m_b being of similar amplitude as m_c . An observed value of $P/P_0 = -1$ for different magnetic reflections at different omega angles simply states the absence of NSF scattering and in consequence that the magnetic moments lie in the b - c plane (taking into account the experimental geometry), but it cannot be concluded that m_b is similar to m_c . In fact, any moment distribution in the b - c plane would yield $P/P_0 = -1$ for all (0KL) reflections with finite intensity.

We thank the Referee for urging us to clarify the phrasing in caption of Fig. “Neutron flipping ratio in sample A for two components of the magnetic order in DyTe₃'s ground state” in Extended Data. We have revised the caption as follows:

“In our geometry, SF intensity is dominated by the b -component (the c -component) of the magnetization at $\omega \approx 0^\circ$ (at $\omega \approx 90^\circ$), although the structure and magnetic form factors have to be carefully taken into account when comparing intensities of various reflections (section E5). **a**, The incommensurate magnetization component at \mathbf{q}_{cyc} gives dominant SF scattering, with $P/P_0 \equiv -1$ independent of the ω angle. This implies presence of both m_b and m_c for the incommensurate reflection.”

3. The integrated intensities have been analysed by including two magnetic domains. Apart from the spin components a phase angle δ was refined for the cycloidal part, which represents the phase shift between the upper and lower cycloids of a zig-zag chain. Using the nomenclature of Dy_j in Fig. 3g, Dy₂ is displaced by $z = 1/2$ from Dy₁ and therefore the angle between those two spins, for $\delta = 0$, would be $\mathbf{q}_{\text{cyc}} \cdot \mathbf{z} = 0.21 \cdot 1/2$ in fractions of 2π , i.e. 37.8° . A reported phase angle of 58° would therefore mean an angle between the Dy₁ and Dy₂ spins of $37.8^\circ + 58^\circ = 95.8^\circ$ which corresponds well to Fig. 3g and Fig. E7e (on the other hand, in line 556 a phase angle of $\delta \sim 2\pi/5 = 72$ degrees is reported, is $58^\circ \sim 72^\circ??$, 58° is rather $2\pi/6$).

We apologize for several typesetting errors regarding the δ -angle in the original manuscript, which arose due to miscommunication between the authors. Firstly, we note that the correct value of δ for two domains in Sample A was refined to ± 73 degrees, not ± 58 degrees, for the dataset of sample A. The incorrect value in the previous version originated from miscommunication between the authors (the refinement is unchanged). Moreover, the δ for Sample B is given as ± 79 degrees in the revised Extended Material (section “Results of Magnetic Structure Analysis”). Thus, the two samples yield reasonable agreement regarding this matter.

Following the Referee’s comments, we have revised the manuscript and added a figure to the Extended Data, explaining the definition of the δ -angle for the case of co-propagating cycloids (Figure: “Two ‘ δ -domains’ (A’ and B’), consistent with the $\mathbf{q}_{\text{AFM}} = (0, b^*, 0.5c^*)$ ordering vector, for the incommensurate part of the magnetic order in phase I of DyTe₃”).

Fig. R2 shows the previous and revised versions of the illustration regarding the definition of δ . We have also added the section “Symmetry and structure factor: commensurate component in phase I”, which discusses the definition of δ in more detail.

Fig. R2: Revised (upper) and previous (lower) illustration of cycloidal component of the magnetic order in DyTe_3 , for two domains of the phase angle δ . The revised illustration can be found in Fig. “Two ‘ δ -domains’ (A’ and B’), consistent with the $\mathbf{q}_{\text{AFM}} = (0, b^*, 0.5c^*)$ ordering vector, for the incommensurate part of the magnetic order in phase I of DyTe_3 ”, which was added to the Extended Materials.

In line 549 it is mentioned that a single parameter $\pm \delta$ is refined for both domains, even though in the caption of Fig. E7e it is described that the phase angle was refined individually in both domains. There is a lack of consistence here.

We thank the Referee for finding this inconsistency in the text. In fact, a single value of the δ angle was refined, and domain A’ and B’ correspond to $\pm \delta$, respectively. As discussed in section “Symmetry consideration (incommensurate)” in the revised Extended Data, the two domains of δ are related (for example) by a mirror operation.

The former figure E7 was broken into several parts in the revision, and the error in the caption was removed. In the revised manuscript, Fig. “Two ‘ δ -domains’ (A’ and B’), consistent with the $\mathbf{q}_{\text{AFM}} = (0, b^*, 0.5c^*)$ ordering vector, for the incommensurate part of the magnetic order in phase I of DyTe_3 ” provides a clear explanation of the δ -angle and the corresponding two domains for the incommensurate order.

I will assume that δ has the same value but different sign in the two domains, but then the domain with $-\delta$ does not agree with the magnetic structure shown in Fig. E7d. The spin of Dy_2 seems to have an angle of -58° with Dy_1 , instead of $37.8^\circ - 58^\circ = -20.2^\circ$. This is again inconsistent.

Again, we apologize for inconsistencies in the previous version of the manuscript. As noted in the previous paragraphs, we have made several changes and corrections to the manuscript in response to the Referee's comments. The figure noted here was also corrected (see Fig. R2).

Assuming that just the picture is wrong, I reproduced the magnetic structures in both domains (using FullProf and Mag2Pol, two freely available programs to analyse diffraction data) and obtain that the intensity of the $(0\ 1\ 1+q_{cyc})$ reflection is about 2 times stronger than that of the $(0\ 9\ q_{cyc})$ reflection (including the Lorentz factor). If I use the presumably wrongly depicted magnetic structure in Fig. E7d (which does not have any symmetry relation to the first domain), the $(0\ 1\ 1+q_{cyc})$ reflection would be roughly 50% stronger in intensity than the $(0\ 9\ q_{cyc})$ reflection. This does not agree with the raw data shown in Fig. 3e,f (polarised neutrons) and shows the consequence of its misinterpretation (see my previous point). Yet, the model apparently explains the unpolarised neutron data very well (Fig. E7c), which constitutes a serious problem of consistency and sheds doubts on the validity of the results. In the same line, why didn't the authors comment on the different intensities of the commensurate reflections shown in Fig. 3a,b?

The Referee has carefully examined the details of the data in the main text, and is asking us to compare the observed polarized neutron data to calculated intensities, which was not done in the previous version of the paper. We think the question is best answered by an analytical model for the scattered intensity, so that there can be no confusion about the software and model parameters used.

We have largely expanded the Extended Data and derived analytic expressions for the neutron scattering cross-section of \mathbf{q}_{AFM} and \mathbf{q}_{cyc} of $DyTe_3$ assuming the existence of equally populated domains, in sections "Structure factor calculation (incommensurate)" and "Structure factor calculation (commensurate)". Further, in section "Example: intensity ratio of two reflections" of the revised Extended Data, we directly address the question raised by the Referee.

The refinement of incommensurate magnetic reflections is based on the ansatz

$$\mathbf{m}_j = 2\mathbf{X} \cdot \cos(\mathbf{q} \cdot \mathbf{r}_j + \varphi_j) - 2\mathbf{Y} \cdot \sin(\mathbf{q} \cdot \mathbf{r}_j + \varphi_j)$$

with two orthogonal vectors \mathbf{X} and \mathbf{Y} of length X and Y , respectively. Based on this model, we find that a nearly undistorted cycloid ($Y/X \sim 1$) well describes the intensity ratio of the two reflections in Fig. 3, $(0\ b^* \ c^*+q_{cyc})$ and $(0\ 9b^* \ q_{cyc})$. This is consistent with the refinement of unpolarized neutron diffraction data of sample A in the previous version of the manuscript.

However, for sample B with larger data set and absorption correction, the refinement shows a sizable distortion of the cycloid. In the Extended Data, we have added the following discussion regarding this problem:

“Table E2 shows the numerical values for various steps in the calculation. From the observed ratio of peak intensities in Fig. 3, $r=0.77$ and $\gamma = Y / X \sim 0.97$ for these two reflections measured on sample A. This γ is somewhat larger than the result for sample B with full refinement, but sample A is a larger crystal, with anisotropic shape, where absorption correction was not applied. In particular, the observed intensities are expected to be larger along $(0, 1, L)$, close to transmission geometry. Such limitations of the data quality for sample A do not affect polarization analysis and the qualitative evolution of line scan intensities with temperature.”

4. As pointed out in my previous point, there may be some errors in the authors' calculations of structure factors or the data are unreliable and compatible with different magnetic structure models. Did the authors use their proper code for the calculations and refinements? They do not mention any software package known in the neutron diffraction community.

We appreciate the Referee's careful check of our magnetic structure calculations. As noted in the answer to question 3, we have derived analytic expression for the structure factors of various spin textures in DyTe_3 , to clarify potential misunderstandings about the results of the structure refinement. We also note that the revised version – although the data analysis was improved by absorption correction, improved background subtraction, and so on – we use the same approach to magnetic structure refinement.

Therefore, we assert that there are no errors in the calculation of the structure factors, although there were errors in the manuscript text due to miscommunication between members of the collaboration. We have corrected these, following comments from the Referee. For example, there were indeed problems with the definition of the δ angle in various figures of the Extended Data, and an incorrect number was given for δ in the text. We thank the Referee for helping us to clarify these issues.

In the previous version, and in the revised manuscript, we use our own code to calculate the magnetic structure factor. To carefully address the Referee's concern, two members of the collaboration independently reproduced the same results for the refinement of sample A (Fig. R1), confirming the results of the previous version of the manuscript. We have also compared the results of our code to calculation of structure factors using analytic expression under the assumption of two equally populated domains, and obtained consistent results.

But apart from that, there is another major problem with the data analysis. The phase angle δ is not a refinable parameter as the Dy ions in the 4 different sheets (labelled $j = 1$ to 4) are related by symmetry. The little group of space group Cmcm with a propagation vector $(0\ 1\ 0.21)$ contains 4 symmetry operators in the $(000)^+$ set, which are the identity, a 2-fold screw axis in $0,0,z$, a glide plane c in $x,0,z$ and a mirror plane in $0,y,z$. These 4 symmetry operators together with the coset

generated by the C-centering generate the 4 Dy positions in the conventional unit cell. The atoms Dy₂ and Dy₄ are related to Dy₁ and Dy₃ by both the 2-fold screw axis and the glide plane *c* (due to the special position 4*c* with *x* = 0), which invert the rotation sense of the cycloid. The cycloids on sheets *j*=2 and *j*=4 rotate therefore in the opposite sense in comparison to those on *j*=1 and *j*=3. In consequence, the magnetic structure - respecting the underlying nuclear symmetry - cannot induce a polarisation along the *b* axis as claimed in the caption of Fig. 1c and does not possess a fixed sense of rotation as indicated in line 65.

We thank the Referee for detailed questions regarding structure symmetry analysis here, and in the following comments. We have added new sections to the Extended Data [“Symmetry consideration (commensurate)” and “Symmetry consideration (incommensurate)”] where the points raised here by the Referee are discussed at length.

In particular, we consider symmetry lowering starting from the *Cmcm* space group in two steps: First, symmetry reduction due to commensurate AFM order with magnetic space group *C_c2/m*, and subsequently the insertion of a cycloid into a single layer, which breaks further symmetries. We exclude a higher-symmetry (average space group *C2/c*) order with counter-propagating cycloids based by comparison of calculated and observed structure factors, and arrive at average magnetic space group *Cc*, where δ becomes an adjustable parameter and both co-propagating and counter-propagating cycloids are possible in principle. In section “Magnetic structure model (incommensurate)”, we discuss how only co-propagating cycloids can optimize their (next-nearest neighbor or inter-chain) energy by adjusting δ away from 38 degrees, while counter-propagating cycloids can benefit from local Dzyaloshinskii-Moriya interactions (DMI). Several relevant models for counter-propagating cycloids are ruled out based on the observed neutron intensities.

We summarize the discussion in Fig. “Space group *Cmcm* and its symmetries in the (000)+ set, as well as breaking of bond-symmetry by AFM commensurate order”, which has been added to Extended data. Here, we point out that the breaking of symmetry between bonds – which is induced by the AFM *uudd* or *uddu* order – is essential from the viewpoint of symmetry reduction.

We thank the referee for raising these points, which have helped us to improve the manuscript.

The derivation of complex magnetic structures requires an in-depth symmetry analysis. The two domains mentioned by the authors result from a loss of a symmetry operator at the transition into the magnetically ordered state.

As the Referee points out, the emergence of magnetic order breaks some symmetry operations in many magnetic materials. The simplest cases are cubic Fe, Co, and Ni, which undergo the ferromagnetic transition with the structural transition from cubic to tetragonal / rhombohedral, or even lower symmetry. As for helimagnets, there are two scenarios: In one class of magnets, non-collinear/non-coplanar magnetic order originates from the antisymmetric exchange interaction (Dzyaloshinskii-Moriya interaction). Then, the magnetic structure is mostly determined by the crystallographic symmetry. In another

class of magnets, frustration – in a broad sense – may induce non-collinear / non-coplanar magnetic order. Here, the magnetic order tends to break the crystallographic symmetry.

Some typical cases can be found in the material class of multiferroics (spin-driven ferroelectrics), where symmetry breaking by a single-sense helical magnetic order is rather widely observed in 'zigzag-chain' magnets. These have a chain of magnetic sites composed of two sublattices, connected by space inversion, a twofold screw along the zigzag chain, and / or a glide plane between the two sublattices. MnWO_4 is a typical zigzag chain helical magnet with two Mn sublattices of fixed rotation sense. $\text{Ni}_3\text{V}_2\text{O}_8$ also hosts a single-sense helix, while it has a more complicated crystal structure with more than two sublattices. Another famous multiferroic, CuO , also exhibits helical magnetism with two sublattices in a primitive unit cell, although the structure is not of the zigzag type. In CuO , the *c*-glide plane is broken by the helical magnetic order.

We have added a number of references to the discussion section of the main text, to put our work in perspective:

"Symmetry breaking with cycloid / spiral magnetic order of fixed helicity is rather widely observed in zigzag chain magnets (Extended Data, section E4A)"

And in Extended Data:

"...the spontaneous formation of helimagnetism with a unique sense of rotation is common in zigzag chain magnets, and more generally in systems where two or more sublattices are connected by space inversion, a twofold screw axis along the chain, and / or a glide mirror. Examples are $\text{Ni}_3\text{V}_2\text{O}_8$, CuO , and the zigzag chain magnet MnWO_4 , which all host single-sense helices under these conditions."

The description of the magnetic symmetry - at least using irreducible representations, but ideally using magnetic space groups and superspace groups, is mandatory. In fact, the 2 domains of the q_{AFM} structure can be described by the two basis vectors of a two-dimensional irreducible representation or alternatively by the two magnetic space groups C_c2/m ($-b+c$, a , c ; 0 , 0 , $1/4$) and C_c2/m ($b+c$, $-a$, c ; 0 , 0 , $1/2$) (note the different basis transformations for the two domains).

We thank the Referee for these constructive comments. Using the (black-and-white-type) Belov-Neronova-Smirnova notation, the commensurate antiferromagnetic order is described by the magnetic space group C_c2/m (monoclinic), as the Referee points out. We note a minor typo in the Referee's comment regarding the sign of the basis vectors – the ones provided do not form a right-hand frame.

We have added a suitable discussion of magnetic space groups for the commensurate components to the section "Symmetry consideration (commensurate)", and we have added a figure "Unit cell, magnetic domains for commensurate spin component, and illustration for structure factor calculation in phase I of DyTe_3 (antiferromagnetic part, AFM)" to the Extended Data. This figure shows that *uudd* and *uddu* domains are distinguished by the monoclinic tilt in magnetic space group C_c2/m .

The q_{cyc} structure follows a one-dimensional irreducible representation or a magnetic superspace group, which yields of course a different spin configuration than the one presented by the authors. The authors have not considered group theory aspects and present a solution which cannot be justified based on the data at hand, i.e. without further proof of symmetry reduction.

In summary of our response to question 4 by the Referee, we have added a large body of additional material, figures, and discussion to the Extended Data to discuss the symmetry lowering of the average space group step by step, which can approximate for the incommensurate order parameter q_{cyc} . We discuss the lowering from $Cmcm$ to C_c2/m , and further to $C2/c$ or Cc when inserting a cycloid of fixed helicity in a single Dy-layer of the crystal structure.

Using the abovementioned magnetic space group C_c2/m , the incommensurate modulation vector may be described as $(-q, 0, q)$. However, such a change in the cell vector orientations across the magnetic transition may be confusing many readers. We hence describe the incommensurate magnetic order based on the crystallographic $Cmcm$ unit cell, without considering the effect of magnetic order or the CDW on the unit vectors.

Finally, as we do not have any information about the phase relation between the CDW and incommensurate magnetic order, which makes us hesitate to present the magnetic superspace group based on the available data. We kindly ask for the Referee's understanding about the limitation of scattering techniques regarding this issue.

5. The attribution of J_1 and J_2 to the nearest-neighbour (NN) and next-nearest-neighbour (NNN) interactions is in principle very simple, but there are several inconsistencies throughout the manuscript. In line 58 it is mentioned that the NN is located in the respective other layer. While this seems to match with the perspective sketches of the zig-zag chains, it is not what Fig. 1a,b suggest and not what can be derived based on the atomic parameters of either the average $Cmcm$ structure or those of the superspace group $C2cm(00g)000$ from Ref. 18. The distance corresponding to J_1 (equivalent to the c lattice parameter) is shorter than that corresponding to J_2 (c.f. Fig. 1b).

Firstly, we have corrected the label of Fig. 1b and have interchanged the labels of J_1 and J_2 . To be clear, J_1 should be the exchange between two square net layers, and we thank the Referee for catching the mistake in the illustration.

There is an issue here, because we would like to define J_1 and J_2 as suitable for the zigzag chain model, although these do not correspond to the *structural* nearest- and next-nearest neighbor interactions in the case of DyTe_3 , in terms of absolute distance of ions. In our view, this is a problem of semantics, but it is true that the previous manuscript version used the term "nearest neighbor" (and so on) too loosely.

In the revised version, we retain the notation " $J_1 \rightarrow$ structural next-nearest neighbour exchange but nearest neighbor on the zigzag chain", and so on. This convention is

reasonable from the viewpoint of the model, and in typical zigzag chain systems, J_2 is required to be larger than J_1 to stabilize the $uudd$ order [Phys. Rev. B **75**, 064413 (2007)].

We have changed the definitions in the text and carefully re-examined all references to J_1 and J_2 as instructed by the Referee. In the text, we have also removed many references to “nearest-neighbour” or “next-nearest neighbour” exchange, to avoid confusing the reader. A main point of change is the modified caption in Fig. 1, which reads as follows in the revised version:

“Zigzag chain illustration of double-square net structure in DyTe_3 . The interactions J_1 , J_2 connect nearest and next-nearest neighbors in the zigzag chain model, respectively; but the inter-atomic distance in the crystallographic structure is shorter for J_2 , and its antiferromagnetic coupling strength is dominant.”

In Extended data, we also write more clearly, for a spin-model calculation:

“...where n labels magnetic moments on a single layer of a single zigzag chain. Here, n , $n+1$ represent nearest neighbours in the crystal lattice of DyTe_3 , so that their coupling J_2 can be expected to be stronger than the coupling J_1 between the sheets (see next section).

In this respect, the authors mention that they have used the structural parameters from Ref. 18 to calculate the structure factors (line 524), but they do not mention the superspace group formalism or the determined superspace group. The atomic positions in Ref. 18 are in fact incompatible with space group $Cmcm$ (e.g. Dy at position 0.94 0.169803(12) 0.25, note the non-zero x component), and therefore the citation of Ref. 51 (Squires, line 526) is inappropriate and inconsistent as it does not treat the superspace group formalism.

We apologize for the inconsistency in the text. Although we used the \mathbf{q} -vector values from Malliakas *et al.*, JACS **128**, 12612 (2006), and we use their average space group symbol $C2cm$ in chapters “Symmetry consideration (commensurate)” and “Symmetry consideration (incommensurate)” of the revised Extended Data, the internal (fractional) coordinates of the older work V. K. Slovyanskikh *et al.*, Russian Journal of Inorganic Chemistry **30**, 1666-1669 (1985) were sufficient to describe our structural neutron scattering data. These authors reported space group $Cmcm$, neglecting the effect of the CDW on average atomic positions. Specifically, our crystal structure analysis In Fig. “Nuclear reflections observed in neutron scattering, consistent with orthorhombic crystal structure of DyTe_3 ” of Extended does not consider the incommensurate CDW distortion via the superspace group formalism, as the Referee states correctly.

The section discussing nuclear scattering was expanded in the revised manuscript and moved to Extended Data. To correct the issue pointed out by the Referee, we have added the following statement there:

“We find good agreement of the experimental scattering data and model when using the atomic positions from the high-temperature $Cmcm$ space group of DyTe_3 [76]. In reality, the formation of charge order below $T_{CDW} \approx 320$ K lowers the symmetry, as discussed by Malliakas *et al.* in”

Also, the authors should have discussed how the structural superspace group $C2cm(00g)000$ - being polar along the a axis - can be in agreement with a supposedly polar distortion along the b axis (see caption of Fig. 1c) via the reported magnetic structure. The manuscript lacks a decisive point here, as no symmetry analysis was carried out.

We thank the Referee for raising these issues, again regarding symmetry analysis. As noted above, we have added to the revised manuscript two sections, “Symmetry consideration (commensurate)” and “Symmetry consideration (incommensurate)”, as well as several figures regarding symmetry analysis.

Here, we discuss symmetry lowering from the $Cmcm$ space group via antiferromagnetic correlations (commensurate) and subsequently, via incommensurate correlations driven by the charge-density wave order. We point out that, in large part, the discussion is not significantly changed when moving from average space group $Cmcm$ to $C2cm$. In the former section (commensurate), the magnetic space group of the AFM order is lowered from C_{c2}/m to C_{c2} , but – as the moment direction has been determined from polarized neutron scattering – this does not affect the spin arrangement nor the nature of magnetic domains. In the latter section (incommensurate), the key issue is the breaking of C_{2a} symmetry; the discussion of this section is also broadly unaffected by a potential polar distortion of the CDW state.

We added a statement to the Extended Data:

“Note: A previous x-ray scattering study reports the superspace group $C2cm(00\gamma)000$ for the CDW state in $DyTe_3$. In average space group $C2cm$ (No. 40), the M_a mirror of $Cmcm$ is already broken. Starting from this lower-symmetry symbol, analogous discussion leads to C_{c2} for the commensurate component of the magnetic order in phase I.”

“For $\mathbf{h} = (1, -1, 1, -1)$, we can further lower symmetry to Cc by shifting δ away from δ_0 , thus breaking C_{2a} . However, the spontaneous formation of helimagnetism with a unique sense of rotation is common in zigzag chain magnets [...] Thus, we consider the cycloid of uniform helicity $\mathbf{h}^{(1)} = (1, 1, 1, 1)$ (average space group Cc), which allows the spin system to adjust δ to minimize inter-chain exchange energy. We are left with two helicity-domains, $\mathbf{h}^{(1)}$ and $\mathbf{h}^{(2)} = -\mathbf{h}^{(1)}$ for each AFM domain ($uudd$ or $uddu$). These are related by C_{2a} , which reverses the rotation sense of the cycloids, but maintains the same AFM domain and the same phase relation δ for a given bond (Fig. “Symmetry considerations regarding incommensurate magnetic order in phase I, for $DyTe_3$ ”). As the symmetry is reduced to Cc in each domain, there is no constraint on the number value of δ . The discussion remains qualitatively unchanged if the commensurate part has magnetic space group C_{c2} .”

More directly answering the Referee’s comment, the formation of a spin-cycloid with a fixed sense of rotation requires spontaneous symmetry breaking, and is – in this sense – ‘not consistent’ with the abovementioned $C2cm$ space group. However, as discussed in an answer to a previous question, incommensurate magnetic order is not always consistent with the base lattice symmetry. In the revised manuscript, we have discussed

symmetry lowering and the requirements of spontaneous symmetry breaking when adding a cycloid even in a single layer of the DyTe₃ structure.

We have also added a section describing a more suitable Hamiltonian for magnetic structure modeling (section “Spin Hamiltonian” in Extended Data). Here, an important aspect is the *local* (not global) mirror symmetry breaking by the CDW distortion, to allow terms such as $S_n^a S_n^c$ for a spin \mathbf{S}_n on a one-dimensional chain (site n). Going beyond the model in the previous version of this work, this model not only explains the coexistence of \mathbf{q}_{AFM} and $\mathbf{q}_{\text{cyc}} = \mathbf{q}_{\text{AFM}} \pm \mathbf{q}_{\text{CDW}}$ reflections, but also explains:

- (a) The spin direction in the ground state, via the abovementioned anisotropic exchange terms.
- (b) A transition to magnetic order of purely \mathbf{q}_{CDW} character (suppression of the AFM part) in an applied magnetic field.

The latter point was also observed in our additional small angle neutron scattering experiments.

In summary of this response, we reiterate that the co-propagating cycloid is not preferred by Dzyaloshinskii-Moriya interactions in DyTe₃, but favored by exchange interactions within a DyTe slab. Therefore, the base lattice symmetry is not the (only) driving factor for symmetry breaking and spin texture formation in this material.

6. The previous point is also an issue with the authors’ J_1 - J_2 model used in their Monte Carlo simulations. In Fig. 1b the NN interaction J_1 is defined between two Dy ions on the same sheet (i.e. same atomic position y), while the NNN interaction J_2 is defined between two sheets with different y -values. In Fig. E2 the authors claim that the in-phase model is favoured by the modulated $J_{1,\text{CDW}}$, but refer to spins of different layers to compute the dot product. This is not consistent with the model and the definition of the exchange constants.

In our response to question 5, we have addressed the issue of defining J_1 and J_2 , which is suitably defined for the zigzag chain model (but may be confusing from the point of view of the crystal structure). We have also corrected the labeling of J_1 and J_2 in Fig. 1 in the revised version.

Again, we thank the Referee for pointing out this error in Fig. 1b.

And I have a further problem of understanding: if J_1 and J_2 are modulated (as defined in lines 162-163) by $q_{\text{CDW}} = 0.29$ (note the different value of 0.3 in line 589 and the mistake in line 164, $0.5 - 0.29 = 0.31$, as well as the use of x and z in the sine term, line 163 and 588, respectively), ..

We thank the Referee for bringing these typos to our attention. We have corrected the manuscript accordingly

...all values between $+J_{1,\text{CDW}}$ and $-J_{1,\text{CDW}}$ will be adopted for J_1 along a long enough chain along the z direction. The argument that the in-phase model is favoured by

J_1 is therefore insufficient since large positive amplitudes (depicted by the grey shaded bars in Fig. E2) will be multiplied with positive and negative J_1 values and therefore cancel out in the energy balance. If the authors meant $J_1 = J_1(0) + J_{1,CDW} \sin(q_{CDW} \cdot z)$ (note that the $J_1(0)$ term is not mentioned in the manuscript), then the interval of J_1 values would be $[J_1(0) - J_{1,CDW}, J_1(0) + J_{1,CDW}]$, but there would still be a problem in my opinion: It has to be noted that the compensation between positive and negative amplitudes of $m_i \cdot m_{i+1}$ (e.g. $m_1 \cdot m_2$ and $m_2 \cdot m_3$ in Fig. E2a) is the same between the two models depicted in Fig. E2. Using the reported values of 6.49 μ_B for the AFM component and 6.18 μ_B for the cycloidal component, the angles between different spin pairs in the conical magnetic structure can be calculated for the in-phase and the out-of-phase models. In the in-phase model, the large positive amplitude originates from nearly parallel spins with angle 13.86° [$\cos(13.86^\circ) = 0.971$], while the negative amplitude comes from spins with an angle of 124.9° [$\cos(124.9^\circ) = -0.572$]. The values for the out-of-phase model are 61.53° [$\cos(61.53^\circ) = 0.477$] and 94.48° [$\cos(94.48^\circ) = -0.078$], respectively. In both cases the sum of the amplitudes is 0.399. It is therefore not clear, why one model would be preferred over the other by J_1 since the energy balance in the Hamiltonian would be the same.

We thank the Referee for pointing out issues with the spin Hamiltonian, which as encouraged us to completely rework and simplify the theoretical model, with the target of not only explaining the coupling of two \mathbf{q} -vectors, but also the separation of spin components by \mathbf{q} -vector and – at least qualitatively – the behavior in a magnetic field. To reduce complexity, the revised model – Eq. (1) in the main text – targets a simple 1D chain of dysprosium moments coupled to the CDW via on-site terms as

$$\mathcal{H} = J_2^{AFM} \sum_n S_n^a S_{n+1}^a - \sum_n [E_{CDW}^{ab} \cos(q_{CDW} z_n) S_n^a S_n^b + E_{CDW}^{ac} \sin(q_{CDW} z_n) S_n^a S_n^c]$$

In the Extended data, the role of the terms in the second sum is summarized by:

“As compared to anisotropic exchange interactions, for example of the type $S_n^c S_{n+1}^a + S_n^a S_{n+1}^c$, the present E_{CDW} terms cannot induce spontaneous magnetic order by themselves, but rather create a ‘parasitic’ spin modulation – driven by the charge-density wave of RTE_3 , R = rare earth – on the back of either AFM order below B_c or of the field-polarized moment above B_c .”

This new model is carefully constructed from the viewpoint of symmetry to explain the experimental results. The new model does not require non-biased numerical simulations, since the magnetic frustration is lifted and we can easily predict the analytic solution in Fourier space. We performed variational calculations assuming a spin ansatz, as described in Methods and as depicted in the revised Fig. 4d. In contrast, the previous model was based on magnetic frustration between nearest-neighbor and next-nearest neighbor exchanges, and it is difficult to predict the solution analytically.

Further, the revised version of Extended Data has been supplemented by a section “Spin Hamiltonian”, which discusses in detail the definition of the spin Hamiltonian, its

ability to reproduce the ground state, and its consistency with the field-induced magnetic phase transition for $\mathbf{B} // c$. It is possible, based on the revised Hamiltonian, to discuss the in-phase or out of phase coupling between various components of the magnetic structure, analytically even. However, we believe that such discussion does not add significantly to the manuscript.

Following the Referee's comment, we have thus removed a figure discussing the in-phase or out-of-phase coupling between the AFM commensurate and cycloidal incommensurate spin components. Instead, we add a remark to Methods:

“Based on scattering techniques, we find it difficult to reveal the phase-shift between CDW and the spin cycloid, and between the antiferromagnetic and incommensurate components of the magnetic order; hence, alternative (out-of-phase) locking between cycloid and antiferromagnetic component is also possible (Fig. E1).”

7. In lines 537 and 550 the authors mention that they have assumed equal domain populations for both the q_{AFM} and q_{cyc} components. While this is of course a reasonable assumption, they should have refined the populations against their unpolarised neutron data in order to demonstrate the robustness of their model. If the cycloidal and the antiferromagnetic components are indeed coupled to an in-phase structure then the domain population should be coupled as well, i.e. similar domain ratios should result for the different magnetic structure components. However, refining the two main axes of the cycloidal envelope (m_b , m_c), an eventual phase angle δ and a domain population approaches or exceeds the reasonable number of parameters for a data set of 11 measured magnetic Bragg peaks.

We thank the Referee for this comment, which could indeed not have been addressed with the previous dataset for sample A. With the revised dataset for sample B (see above for details of the experimental condition), we attempted to refine the domain ratio. This yielded the following parameters:

Relaxing the volume fraction f of domain A for the commensurate (domain A' for the incommensurate part), we obtain $f = 0.4250 \pm 0.0043$ ($f = 0.4993 \pm 0.0044$) with $R = 0.038$ ($R = 0.083$) and $m_a = 5.5415 \pm 0.0174 \mu_B$ ($m_b = 3.8730 \pm 0.0271 \mu_B$, $m_c = 6.5273 \pm 0.0278 \mu_B$). Fixing the volume fraction to $f = 0.5$ gives a somewhat worse quality of fit for the commensurate part, $R = 0.058$, with $m_a = 5.4669 \pm 0.0171 \mu_B$.

We believe that our data is well described by equal domain ratio, and that the further relaxation of the domain ratio parameter may be an over-fit of the data. The result has been added to section “Results of Magnetic Structure Analysis” in Extended Data.

However, the Referee's question touches on the crucial point whether the incommensurate and commensurate parts of the structure are truly coupled to each other (or whether, perhaps, the two types of reflections in scattering could indicate phase separation even in our high-quality single crystals with residual resistivity ratio > 100). To further confirm this point, we have carried out small-angle neutron scattering

measurements (SANS), as shown in Fig. 5 of the revised manuscript, and discussed in Methods.

The $\mathbf{q}_{\text{AFM}} = (0, b^*, 0.5c^*)$ and $\mathbf{q}_{\text{cyc}} = (0, b^*, 0.207c^*)$ reflections both give way to $\mathbf{q}_{\text{inc}}' = (0, 0, 0.293c^*)$ above the critical field $B_c = 0.6$ T. This behaviour is reasonably well explained by the Hamiltonian model in the revised manuscript. Considering the scenario of coupled AFM and cycloidal magnetic structure components, we have added to the Extended Data a figure “Coupling of commensurate and incommensurate magnetic order in DyTe₃, tracked by small-angle neutron scattering (SANS) in a magnetic field along the c -axis”. This figure is reprinted below as Fig. R3.

Fig. R3: Coupled incommensurate and commensurate order parameters in DyTe₃. Integrated intensity of commensurate ($\mathbf{q} = \mathbf{q}_{\text{AFM}}$) and incommensurate ($\mathbf{q} = \mathbf{q}_{\text{cyc}}$) reflections of type $(0, -1, -L)$ as a function of magnetic field and at a temperature of $T = 2$ K. The observed intensity of both drops simultaneously at around $\mu_0 H = 0.4$ T, with a fixed intensity ratio $I(\mathbf{q}_{\text{AFM}}) / I(\mathbf{q}_{\text{cyc}}) \sim 1.5$, as indicated by a grey dashed line in panel b. The error bars correspond to Poisson counting errors of the integrated neutron scattering intensity.

8. Fig. E8 shows the temperature dependence of line scans in reciprocal space both using polarised and unpolarised neutrons. Fig. E8a (unpolarised) shows how the q_{cyc} peak position shifts from phase I to phase II, and there is a coexistence of both peaks at $T = 3.7$ K, while only one peak is observable at $T = 3.8$ K. Why is there a signature of both peaks in the SF scattering data shown in Fig. E8b at 3.8 K?

We thank the Referee for pointing out this weakness of the data in phase II. We have considered carefully among all collaborators and finally decided to remove the polarized neutron scattering data in phase II from the manuscript, as well as the structure cartoons for phase II (please see next question). There may be some issue with the temperature control in the polarized neutron scattering experiment in phase II, where an uncertainty of 0.1 K can appear. Although we believe the structure model for phase II is reasonable, and that the data is consistent with the magnetic order depicted, we concur with the Referee that the experimental evidence from PNS is too weak and should not be over-interpreted.

We hope the Referee and Editor will accept this change to the manuscript, which does not affect the core claim of our work. Moreover, we maintain the nonpolarized neutron scattering data (temperature dependence) in the manuscript, as before. These data are well consistent with the thermodynamic phase diagram. Figures “Commensurate antiferromagnetic component in phase II of DyTe₃” and “Incommensurate component in phase II of DyTe₃” in Extended Data have been combined into one, and have been updated with more quantitative analysis of the temperature dependence of the q -position.

Furthermore, the authors do not present supporting data and analysis for the magnetic structure in phase II depicted in Fig. E8d. The polarised neutron line scans in Fig. E8b,c indicate the reduction of the m_c component, but without a refinement to integrated intensity data it is impossible to deduce the phase angle δ which the authors apparently consider to be constant throughout the phase transition. There is no reason to assume this, and there is little information about how this structure was deduced.

We thank the Referee for inquiring about phase II, which – although interesting to us – is of course not the main focus of the research paper. As noted in the response to the previous question, we have re-examined the available polarized neutron data, and removed the PNS data from “Incommensurate component in phase II of DyTe₃ (sample A)” and “Commensurate antiferromagnetic component in phase II of DyTe₃ (sample A)”. We maintain the non-polarized neutron scattering data and have also included a temperature evolution of peak splitting in phase II.

In the main text, following the Referee’s comment, we have modified the discussion of phase II, replacing the former discussion of this phase with the following text:

“Heating the sample above $T_{N1} = 3.6$ K, we observe a peak splitting of q_{AFM} , and a concomitant shift in q_{cyc} that indicates the sustained coupling of the two ordering vectors, via the CDW, at elevated temperatures. The sharp enhancement of χ_c in Fig. 1a further suggests that m_a , m_b survive to higher temperature than m_c , consistent with a putative incommensurate, fan-like order in phase II, which warrants further study.”

Minor details, but making the review more difficult, are the following inconsistencies in figure referencing:

Caption Fig. 2e: There is no panel h.

We removed this comment.

Caption Fig. 2f: Panel d does not show a line scan.

We changed the reference to Fig. 2e.

Beginning of caption Fig. 3: Fig. 2g does not exist

We changed the text as follows: “In the geometry of Fig. 2d...”

Line 145-146: Figs. E6 and E7 refer to the ground state and not to phase II

This part of the main text has been removed, as the polarization analysis for phase II has been removed.

The caption of Fig. 4d is not sufficiently descriptive

We have modified Fig. 4d (now Fig. 4c) based on the revised model calculations specified above, and have added a new, more descriptive, caption:

“Squared moment amplitudes $I_{AFM} = |S^a(q = 0.5 c^*)|^2$ (blue) and $I_{cyc} = |S^c(q = 0.5 c^* \pm q_{CDW})|^2 + |S^b(q = 0.5 c^* \pm q_{CDW})|^2$ (red) from model calculations according to Eq. (1) as functions of E_{CDW}/J_2^{AFM} where $E_{CDW} = E^{ab}_{CDW} = E^{ac}_{CDW}$; five and ten percent threshold for the incommensurate part indicated by a dashed line (Methods).”

Line 84: Fig. 2c is not a thermodynamic or transport probe

We changed the text:

“We characterize the phase transition in DyTe₃ using thermodynamic and transport probes in Fig. 2 a,b.”

Given the limited neutron data quality, the numerous inconsistencies in the neutron data analysis, to some extent wrong conclusions and the absolute lack of symmetry considerations, it is difficult to trust the validity of the presented results concerning the ground state magnetic structure in DyTe₃, which is the key result of this study. Under these circumstances I am sceptical that even a heavily revised version of the manuscript can be considered for publication in Nature Communications.

Again, we thank the Referee for extensive comments, which we aimed to address in detail in our answers above. We also note that not only the structure refinement (in the SI), but also the observation of coupled antiferromagnetic and CDW orders with magnetic field dependence, the modeling of magnetic anisotropy, the observation of noncoplanar magnetic spin components, and so on, constitute ‘key results of this study’.

We are confident that the revised manuscript, with an extensive amount of changes and additional data in response to both the Referees’ comments, merits publication in Nature Communications.

Additional changes not discussed in the response letter above:

- We harmonized the notation, clearly distinguishing between Miller indices and momentum transfer \mathbf{q} everywhere.
- We adjusted the value of q_{cyc} to 0.207 everywhere.
- A number of coauthors, including their contributions and founding information, have been added to the manuscript.
- We corrected a typo in the lattice parameter, determined by x-ray scattering.
- We enlarged the field range in Fig. 4 a, to include phase V as in Fig. 5.
- We included our SANS results and the details of our new model in the abstract.

Reviewers' Comments:

Reviewer #1:

Remarks to the Author:

The authors have done a very impressive job of responding to the referees. I recommend publication and commend the authors on their hard work.

One minor comment: the Raman work by the Burch group didn't show the CDW was "magnetic". Rather, it was "Axial" and thus has a moment. However, the origin of the dipole (electric/magnetic) is not yet clear, but either way could lead to coupling.

Reviewer #2:

Remarks to the Author:

I wish to congratulate the authors for their impressive revision of their manuscript. All my questions and doubts were taken very seriously and discussed in great detail both in the reply and in the manuscript. Extensive explanations, also using new illustrations and equations, have been added which will be very useful for the readers.

I am very glad to see such an improvement and I can now clearly recommend this manuscript to be published in Nature Communications.

Response letter ‘Noncoplanar helimagnetism in the van-der-Waals magnet DyTe₃’

(S. Akatsuka *et al.*)

We thank the Editor and the two Reviewers for careful reading of our revised manuscript and the finally recommendation to publish our manuscript in Nature Communications. Once again, we appreciate that both Referees have sent us thorough and sincere remarks to the first version of our manuscript, which have helped us to improve it a lot. Below we lay out our point-by-point response to their remaining minor remarks, highlighting the Referees’ text by bold font for clarity.

Response to Reviewer #1:

The authors have done a very impressive job of responding to the referees. I recommend publication and commend the authors on their hard work.

We thank the Reviewer for acknowledging our hard work in preparing this revised manuscript and appreciate the final recommendation for publication.

One minor comment: the Raman work by the Burch group didn't show the CDW was "magnetic". Rather, it was "Axial" and thus has a moment. However, the origin of the dipole (electric/magnetic) is not yet clear, but either way could lead to coupling.

We thank the Reviewer for the clarification, and for pointing out our incorrect phrasing in the manuscript. We adjusted the manuscript accordingly; in particular in the “Charge density wave and magnetic order” subsection of the Results section we changed:

“We now argue that cone-type magnetism in DyTe₃ is realized through (i) ~~a spatial modulation of near-neighbour exchange interactions J_1, J_2 in presence~~ **coupling of the magnetic texture** to charge-density wave (CDW) order and (ii) unconventional single-ion anisotropy.”

And in the Discussion section:

“For example, the CDW’s gapped collective excitation, termed Higgs mode, shows ~~a magnetic an axial~~ **axial** character in RTe₃ as observed via Raman scattering experiments [15], ~~and should thus have a spatially modulated (electric or magnetic) moment. and its~~ **The evolution of this unconventional CDW** below T_{N1} may provide insights on both the origin of magnetic order and the nature of the CDW in DyTe₃.”

Response to Reviewer #2:

I wish to congratulate the authors for their impressive revision of their manuscript. All my questions and doubts were taken very seriously and discussed in great detail both in the reply and in the manuscript. Extensive explanations, also using

new illustrations and equations, have been added which will be very useful for the readers.

I am very glad to see such an improvement and I can now clearly recommend this manuscript to be published in Nature Communications.

Again, we thank the Reviewer for his extensive comments on the first version of our manuscript, which ultimately improved it a lot, and appreciate now the recommendation for publication in Nature Communications.

Additional changes not discussed in the response letter above:

- We have shortened the abstract according to the guidelines.
- We adjusted the terminology from “Extended ...” to “Supplementary ...” in the main text.
- We adjusted the figure and table label in the Supplementary File, and changed the references in the main text according to the guidelines.
- We decoupled the Supplementary File from the Main text and added a separate Supplementary bibliography.
- We adjusted the section structure of the main text according to the guidelines.
- We reworked the main text figures to fulfill the guidelines.
- We corrected the unit labels “a.u.”, “arbit. u”, etc. in all figures to the correct form “arb. units”.
- We added information to error bars in all figure captions, where missing.
- We corrected some minor typos.